# Elucidating ATP's role as solubilizer of biomolecular aggregate

**Susmita Sarkar[1], Saurabh Gupta[2], Chiranjit Mahato[2], Dibyendu Das[2], Jagannath Mondal[1]***

[1]Tata Institute of Fundamental Research Hyderabad, Hyderabad, India; [2]Indian Institute of Science Education and Research Kolkata, Kolkata, India

## eLife assessment

The authors combined molecular dynamics simulations and experiments to study the role of ATP as a hydrotrope of protein aggregates. The topic is of major current interest and thus the study potentially makes an **important** contribution to the community. With the revised version, the level of evidence is considered generally **solid**, although there remains concern regarding the unusually high ATP concentration used in the simulation.

**Abstract** Proteins occurring in significantly high concentrations in cellular environments (over 100 mg/ml) and functioning in crowded cytoplasm, often face the prodigious challenges of aggregation which are the pathological hallmark of aging and are critically responsible for a wide spectrum of rising human diseases. Here, we combine a joint-venture of complementary wet-lab experiment and molecular simulation to discern the potential ability of adenosine triphosphate (ATP) as solubilizer of protein aggregates. We show that ATP prevents both condensation of aggregation-prone intrinsically disordered protein Aβ40 and promotes dissolution of preformed aggregates. Computer simulation links ATP's solubilizing role to its ability to modulate protein's structural plasticity by unwinding protein conformation. We show that ATP is positioned as a superior biological solubilizer of protein aggregates over traditional chemical hydrotropes, potentially holding promises in therapeutic interventions in protein-aggregation-related diseases. Going beyond its conventional activity as energy currency, the amphiphilic nature of ATP enables its protein-specific interaction that would enhance ATP's efficiency in cellular processes.

*For correspondence:
jmondal@tifrh.res.in

**Competing interest:** The authors declare that no competing interests exist.

## Introduction

Adenosine triphosphate (ATP), universally present in all living organisms, is recognized as a fundamental energy currency essential for a myriad of cellular processes. While its traditional role in energy metabolism is well-established, recent investigations have unveiled novel dimensions of ATP's involvement, specifically in protein stabilization and solubilization. Although the canonical functions of ATP are comprehensively understood, its impact on cellular protein homeostasis, particularly in the prevention of aggregation and the maintenance of membraneless organelles, remains a subject of growing interest (*Patel et al., 2017*; *Sridharan et al., 2019*; *Nishizawa et al., 2021*; *Kim et al., 2021*).

Beyond its well-known chemical actions originating from high-energy phosphate bond breaking or allosteric effects during enzymatic catalysis, ATP exhibits contrasting physical biological roles, including the stabilization of soluble proteins (*Sridharan et al., 2019*; *Ou et al., 2021*; *Song, 2021*) and the solubilization of insoluble proteins (*Patel et al., 2017*; *Sridharan et al., 2019*; *He et al., 2020*;

*Aida et al., 2022*; *Pal and Paul, 2020*; *Sarkar and Mondal, 2021*). Proteome-wide profiling analyses emphasize ATP-mediated protein stabilization as a prevalent trait within the ensemble of proteins featuring an ATP recognition site (*Sridharan et al., 2019*). Subsequent independent studies further validate ATP-mediated enhancement of protein stability for specific proteins, such as ubiquitin, malate dehydrogenase (*Ou et al., 2021*), and the TDP-43 RRM domain (*Song, 2021*), all possessing an ATP-binding motif. Intriguingly, these proteins, where ATP functions as a substrate or allosteric modulator, typically require a low concentration of ATP (approximately 500 µM) for its effects (*Sridharan et al., 2019*).

Notably, the cellular cytoplasm maintains a significantly higher concentration of ATP (~5 mM), prompting questions about potential additional purpose for this elevated cytoplasmic ATP concentration beyond its conventional roles as an energy source and in protein stabilization, both typically requiring micromolar concentrations of ATP. As a potential function of the excess ATP concentration within the cell, a substantial influence on cellular protein homeostasis is observed, particularly in preventing protein aggregation (*Patel et al., 2017*; *He et al., 2020*; *Pal and Paul, 2020*; *Hayes et al., 2018*; *Greiner and Glonek, 2021*; *Sarkar et al., 2023b*) and maintaining the integrity of membrane-less organelles (at a millimolar range of ATP concentration). ATP has demonstrated its capability to prevent the formation of protein aggregates, dissolve liquid–liquid phase-separated droplets, and disrupt pathogenic amyloid fibers (*Patel et al., 2017*).

Investigations into ATP's impact on protein solubility and aggregation encompass a diverse array of proteins, including phase-separating proteins like FUS, TAF15, hnRNPA3, and PGL-3 (*Patel et al., 2017*), as well as the aggregates of *Xenopus* oocyte nucleoli (*Hayes et al., 2018*). Elevated ATP concentrations are also observed in metabolically quiescent organs, such as the eye lens, where they prevent the crowding of γS-Crystallin (*He et al., 2020*). Proteome-wide profiling analyses underscore ATP's ability to manage the solubility of numerous proteins, indicating a broad impact on cellular protein solubility and aggregation propensity (*Sridharan et al., 2019*).

Remarkably, a majority of proteins lacking any ATP-binding motif, are reported to undergo solubilization in the presence of ATP. Despite past conformational analyses of water-soluble ATP-binding proteins in the presence of ATP (*Sridharan et al., 2019*), the role of ATP in biomolecular solubilization, specifically how ATP influences the structural plasticity of insoluble, non-ATP-binding proteins, has yet to receive mainstream attention. In this investigation, we employ a combination of computer simulations and experimental techniques to explore how ATP influences the conformational behavior of two proteins situated at opposite ends of the structural spectrum of non-ATP-binding proteins: the folded globular mini-protein Trp-cage and the amyloidogenic intrinsically disordered protein (IDP) Aβ40, associated with Alzheimer's disease (*Figure 1* and *Figure 1—source data 1*). Our study, incorporating Thioflavin T (ThT) assay and transmission electron microscopy, validates hypotheses generated through computer simulations regarding ATP's impact on the nucleating core of the Aβ40 peptide segment. The results establish a clear correlation between ATP's effect on the conformational landscape of these proteins and its ability to impede the aggregation process, affirming its role as a solubilizing agent.

## Results

### ATP's impact on a globular mini-protein

We initiated our investigation via computationally simulating ATP's effect on the archetypal monomeric form of mini-protein Trp-cage which is known to have a well-defined fold under native conditions (see *Figure 1*). The effects of ATP on the structure of the Trp-cage protein were investigated using classical molecular dynamics (MD) simulations. The protein was modeled with a99SB-*disp* force field and solvated with TIP4P-*disp* water molecules. The system was neutralized with chloride ions. The protein was simulated in aqueous solution and also in presence of ATP, at a temperature of 303 K. The precedent studies reporting the effect of ATP on structured proteins, had been performed by maintaining ATP:protein stoichiometric ratio in the range of $0.01 \times 10^3$ to $0.03 \times 10^3$. Likewise, in our simulation with Trp-cage, the ATP:protein ratio of $0.02 \times 10^3$ was maintained. In this study, we opted to maintain the ATP stoichiometry consistent with biological conditions and previous in vitro experiments (*Patel et al., 2017*). Instead of keeping the protein concentration within the micromolar range and ATP concentration at the millimolar level, we chose this approach to avoid the need for an

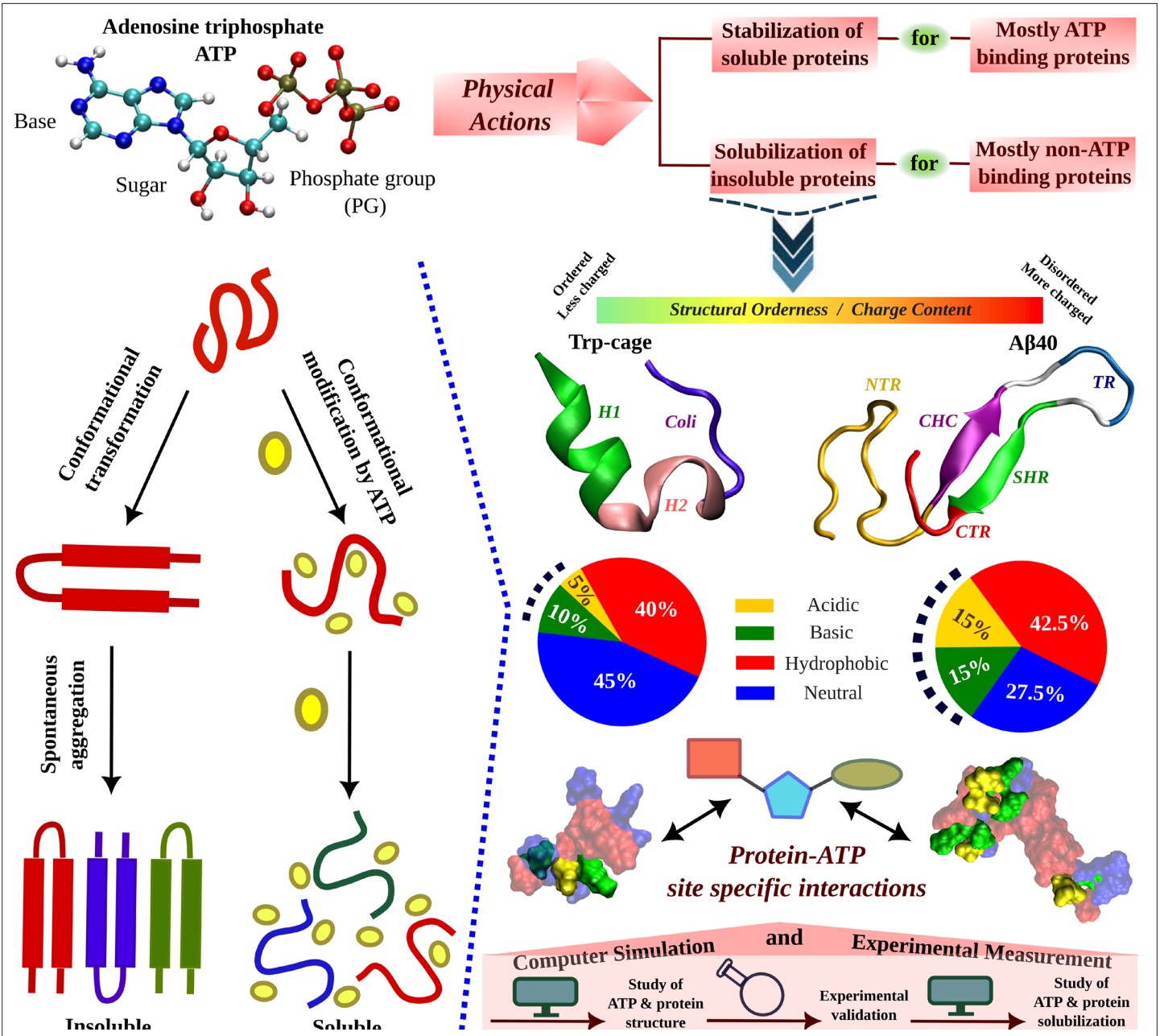

**Figure 1.** Schematic Representation of ATP's Role in Modulating Protein Aggregation and Conformational Plasticity. A schematic representation shows protein undergoes spontaneous aggregation in aqueous medium through specific conformational transformation prone to aggregation. Adenosine triphosphate (ATP) can prevent protein aggregation and improves its solubility. ATP's effect in protein conformational plasticity has been tested for two contrasting protein molecules belonging from two extreme spectrums of the protein family. One is the globular, structurally ordered protein Trp-cage and the other one is intrinsically disordered protein (IDP), Aβ40, containing comparatively more charged residues (according to the nature of typical IDPs). The highly aggregation-prone Aβ40 protein is popularly well known for causing neurodegenerative disorders (*Alzheimer's disease*, *AD*). The structures of both the proteins are shown in the new cartoon representation highlighting the protein region-wise coloration scheme. For Trp-cage the three distinct regions, (1) Helix (H1, 1–9), (2) 3–10 Helix (H2, 10–15), and (3) Coil (coil, 16–20) are shown in green, pink, and navy blue colors, respectively. For Aβ40, the (1) N-terminal region (NTR, 1–16), (2) central hydrophobic core (CHC, 17–21), (3) turn (TR, 24–27), (4) secondary hydrophobic region (SHR, 30–35), and the (5) C-terminal regions (CTR, 36–40) are shown in gold, purple, blue, green, and red colors, respectively. The hydrophobicity index of each of the proteins is shown in pie chart representation containing acidic (gold), basic (green), hydrophobic (red), and neutral (blue) residue content. Each of the proteins shown in the surface model is colored according to the respective hydrophobicity nature. Protein–ATP (base part: red, sugar moiety: cyan, and phosphate group: green) site-specific interactions are tested. The current study of ATP's effect on protein conformational plasticity is performed combining both simulation and experiment based on computational predictions validated by experimental measurement followed by computational reasoning and correlation of ATP-driven conformational modification to protein aggregation scenario.

*Figure 1 continued on next page*

*Figure 1 continued*

The online version of this article includes the following source data for figure 1:

**Source data 1.** Sequence-based analysis.

extremely large simulation box, which would greatly reduce computational efficiency by more than 150-fold.

Simulations were performed at three different concentrations of ATP.Mg$^{+2}$: 0 (aqueous media), 0.1, and 0.5 M. For ensuring exhaustive exploration of protein conformational landscape condition, each condition was extensively simulated via replica exchange MD simulation (see method). The results showed that ATP caused the Trp-cage protein to unfold. This was evident from an increase in the protein's radius of gyration ($R_g$) and a decrease in the number of intra-chain contacts (see *Figure 2A–C*). The $R_g$ of the protein monotonically increased by the action of ATP compared to that of the neat water system in a concentration-dependent manner (*Figure 2A*). The number of overall intra-chain contacts (*Figure 2B*) and native subset of intra-chain contacts (*Figure 2C*) monotonically decreased upon increasing the concentration of ATP from 0.1 to 0.5 M. This is also evident from the inter-residue contact map (see *Figure 2—figure supplement 1A, B*). The two-dimensional free energy landscape of the Trp-cage protein was also calculated (see *Figure 2D–F*). Since multiple previous studies have reported benchmarking of several features of proteins as well as IDPs using both linear and artificial neural network-based dimension reduction techniques and have demonstrated that $R_g$, in combination with fraction of native contact serves as optimum features, we have chosen these two metrics for developing the 2D-free energy profile (*Ahalawat and Mondal, 2018*; *Menon et al., 2024*). The landscape showed that the protein prefers to remain in its folded form (with less $R_g$ and high native contacts) in the absence of ATP. However, in the presence of ATP, the landscape becomes more populated with conformations that have a higher $R_g$ and fewer intra-chain contacts. This suggests that ATP drives the unfolding of the Trp-cage protein by stabilizing conformations that are more extended and disordered. The observation remains robust over the choice of force field. Similar to the results obtained with a99SB-*disp* force field, when the similar simulations were performed with Charmm36 force parameters of protein (with Charmm TIP3P water model) in 0 and 0.5 M of ATP (modeled with Charmm36 force field parameters), the extension of the protein chain has been noted, as was characterized by the stabilization of protein conformations with higher $R_g$ and lower native contacts is attained in presence of ATP (*Figure 2—figure supplement 2*). The event of unfolding of the protein upon addition of ATP is also evident from the simulation snapshots (see *Figure 2G, H*). A comparative molecular analysis of secondary structure of Trp-cage in neat water versus ATP indicates reduction of both alpha-helical and 3–10 helical content (*Figure 3A, B*) along with inter-residue hydrogen bond (*Figure 3—figure supplement 1*). Interestingly, the protein's solvent-accessible surface area (SASA) increases slightly in presence of ATP (*Figure 3C*), implying that ATP can increase the solubility of protein, a point that we would later come back to.

The solvation shell of ATP around Trp-cage suggested its direct interaction with the protein (see *Figure 3D* for a snapshot). Analysis of the preferential interaction coefficient ($\Gamma$) profiles (see method for definition) provided insights into the relative influence of different chemical components of ATP (triphosphate group, sugar moiety, and aromatic base) on the protein (see *Figure 3E*). Consistently higher preferential interaction coefficient ($\Gamma$) profile of the aromatic base indicated that ATP's interaction with the protein is primarily driven by this moiety. This finding is supported by the hydrophobic nature of a significant fraction of Trp-cage residues (see *Figure 1* for a pie chart of residue) and the predominance of van der Waals interactions in the favorable base part of ATP–protein interaction (*Figure 3F*). The preferential interaction analysis also revealed that the base part of ATP interacts more strongly with the helix (H1) region of Trp-cage compared to the 3–10 helix (H2) and coil structures (see *Figure 3G*). Consequently, the prominent interaction with the helical region results in a greater disruption of the helical structure (*Figure 3H*) compared to others (3–10 helical or coil), leading to improved solubilization potential. These results provide valuable insights into the specific interactions between ATP and protein as the key for ATP's actions and also enhance our understanding of the interplay between ATP and protein structure.

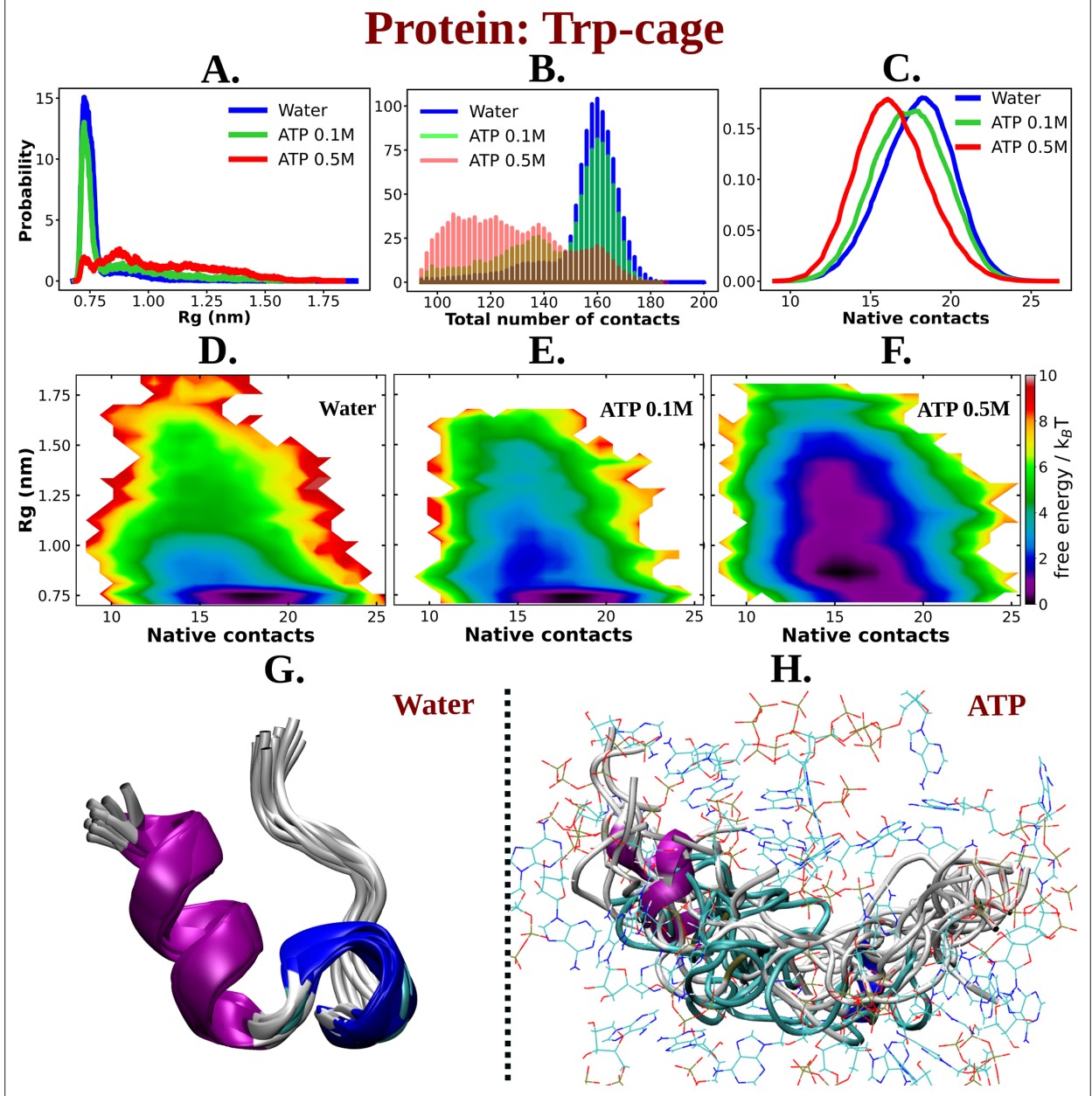

**Figure 2.** ATP facilitates the chain extension of the globular protein Trp-cage. (**A**) The probability distribution of $R_g$ of Trp-cage compared in neat water and in 0.1 and 0.5 M adenosine triphosphate (ATP). (**B**) Probability of the total number of contact formation among the residues of Trp-cage monomer is compared in absence (water) and presence of ATP (at 0.1 and 0.5 M). The probability distribution of native contacts (nc) of the protein in water and in 0.1 and 0.5 M ATP solutions is being shown in (**C**). (**D**), (**E**), and (**F**) represent the 2D-free energy profile of Trp-cage corresponding to the $R_g$ and nc of the protein in water, 0.1, and 0.5 M ATP solutions, respectively. Snapshots containing overlay of protein's conformations in absence of ATP (neat water) and in presence of ATP (0.5 M) are shown in (**G**) and (**H**), respectively. Protein is colored by secondary structure and ATP molecules in (**H**) are shown in line representation with an atom-based coloring scheme (C: cyan, N: blue, O: red).

The online version of this article includes the following figure supplement(s) for figure 2:

**Figure supplement 1.** The residue-wise contact map of Trp-cage monomer is shown for neat water and 0.5 M aqueous adenosine triphosphate (ATP) solution in (**A**) and (**B**), respectively.

**Figure supplement 2.** The 2D-free energy profile of Trp-cage monomer estimated with respect to $R_g$ and native contacts are shown in (**A**) and (**B**) for Trp-cage in neat water and 0.5 M adenosine triphosphate (ATP), respectively, for the simulations with Charmm36 force field.

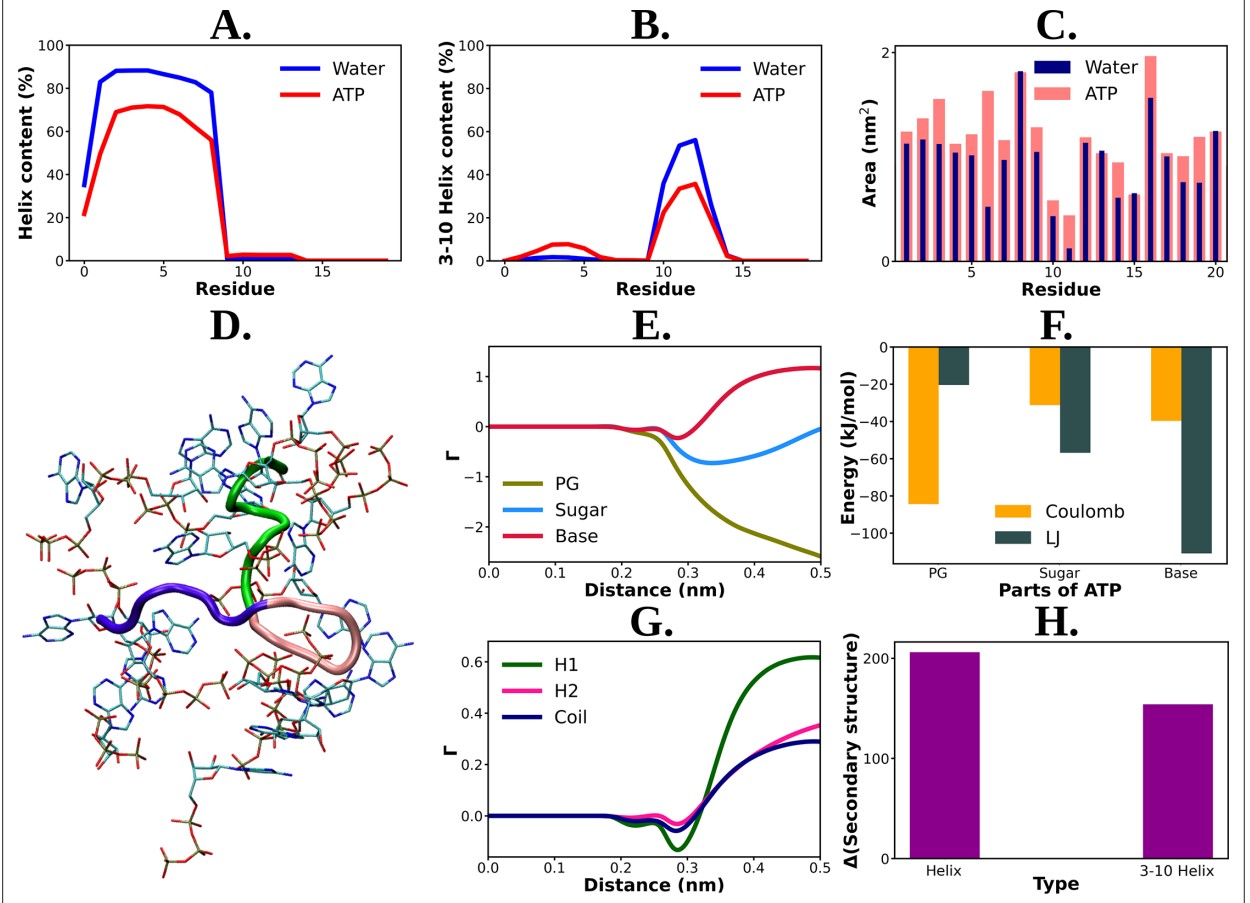

**Figure 3.** Mechanistic understanding of ATP's effect on the conformational modulation of Trp-cage. Residue-wise total percentage of helix and 3–10 helix content of Trp-cage protein in absence and presence of adenosine triphosphate (ATP) (0.5 M ATP) are shown in (**A**) and (**B**), respectively. (**C**) The solvent-accessible surface area is calculated (with gromacs module of 'gmx sasa') for Trp-cage and represented for the aqueous medium without and with ATP corresponding to each of the protein residues in a bar plot representation. (**D**) shows a representative snapshot of ATP's (in licorice representation with atom-based coloring scheme) interaction with Trp-cage (new cartoon representation). Green: H1 (1–9), pink: H2 (10–15), navy blue: coil (16–20). (**E**) Preferential interaction coefficient ($\Gamma$) of different parts of ATP: PG, sugar, and base (with respect to solvent, water) with protein is being compared. (**F**) Bar plot representation of coulombic and LJ interaction performed by all three different parts of ATP (PG, sugar, and base) with Trp-cage. (**G**) represents the comparative plots of the preferential interaction coefficient ($\Gamma$) of ATP with the three structurally different parts (H1, H2, and coil) of Trp-cage. (**H**) The change in the secondary structure content (helix and 3–10 helix) due to action of ATP is being represented. The difference in helix and 3–10 helix content in neat water from that of ATP solution is shown in bar plots.

The online version of this article includes the following figure supplement(s) for figure 3:

**Figure supplement 1.** The probability distribution of the number of intra-chain hydrogen bonds of Trp-cage has been compared for neat water and in 0.5 M adenosine triphosphate (ATP) solution.

## ATP's impact on an IDP

The observation of ATP's unfolding effect on a model globular protein across a range of concentrations encouraged us to explore how its impact would vary on an IDP Aβ40, which sits at the other end of the spectrum of protein family. Aβ40 has also remained a regular subject of pathological neurodegeneration due to its self-aggregation propensity. Accordingly, extensively long MD (30 μs) simulation trajectory of monomeric Aβ40 in 50 mM aqueous NaCl media, as obtained from D. E. Shaw research, was analyzed (*Robustelli et al., 2018*). To realize the effect of ATP on the conformational ensemble of Aβ40, we simulated the conformational repertoire of the monomeric form of Aβ40 in 0.5 M aqueous ATP solution containing 50 mM NaCl salt. The former experiments investigating protein (unstructured) aggregation in presence of ATP, had been performed by maintaining ATP:protein stoichiometric ratio in the range of $0.1 \times 10^3$ to $1.6 \times 10^3$, similarly we have also maintained ATP/protein stoichiometry $0.1 \times 10^3$ in our investigation ATP's effect on disordered protein

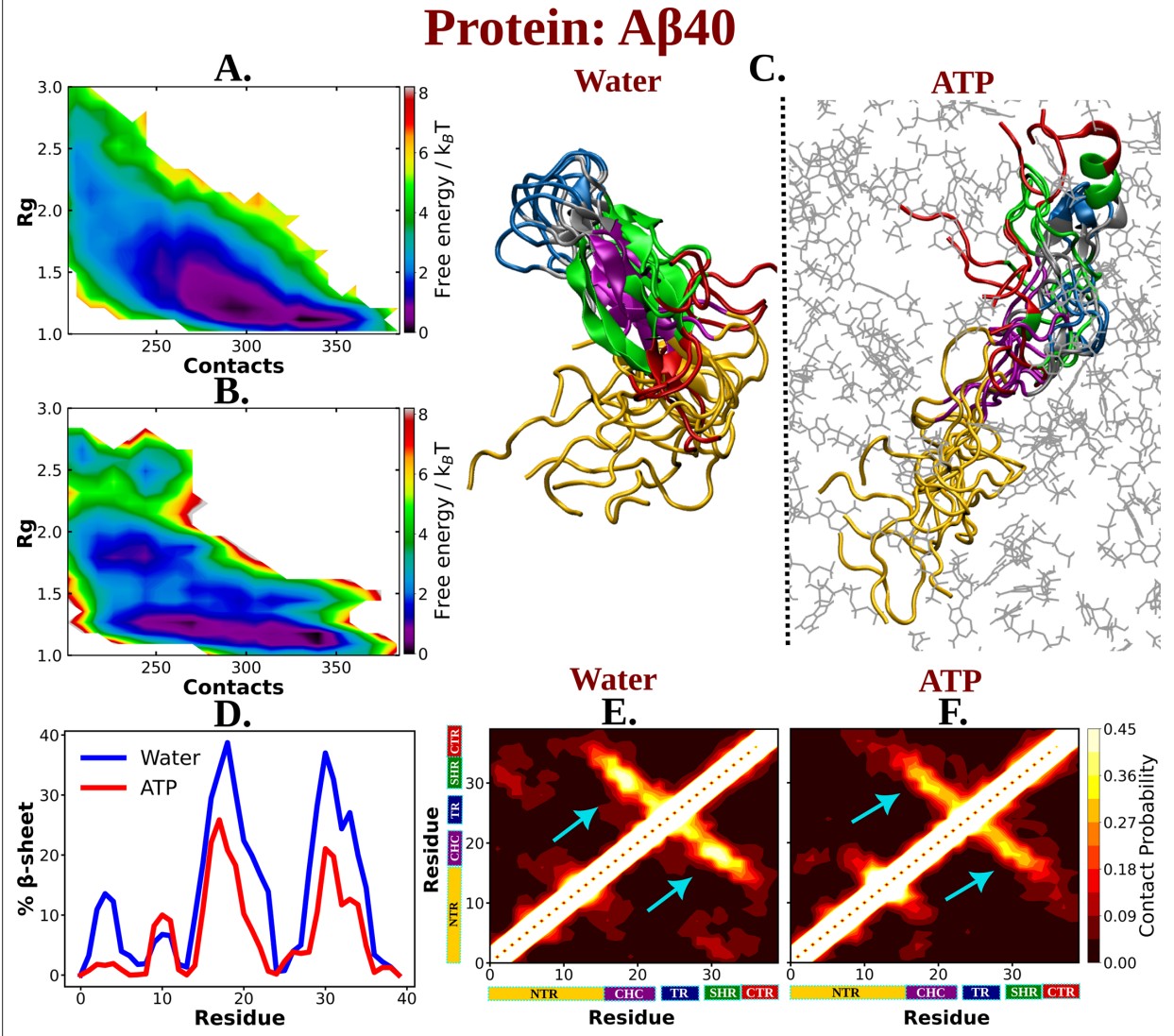

**Figure 4.** The influence of ATP on the protein chain extension of the IDP Aβ40. The 2D-free energy profile of Aβ40 monomer estimated with respect to $R_g$ and total number of intra-chain contacts are shown in (**A**) and (**B**) for Aβ40 in neat water and 0.5 M adenosine triphosphate (ATP), respectively. (**C**) compares the simulation snapshots of Aβ40 monomer in neat water and in presence of ATP. Multiple conformations are overlaid for each of the cases to represent the statistical significance. Protein is colored region-wise as done in **Figure 1** and ATP molecules are shown by gray color line representation. (**D**) compares the β-sheet content of the Aβ40 protein in water and in 0.5 M ATP solution. (**E**) and (**F**) show the residue-wise intra-chain contact map of Aβ40 in absence and in presence of ATP (0.5 M), respectively.

The online version of this article includes the following figure supplement(s) for figure 4:

**Figure supplement 1.** The 2D-free energy profile estimated for Aβ40 protein in absence of adenosine triphosphate (ATP) obtained from replica exchange molecular dynamics (REMD) simulation followed by adaptive sampling simulations.

Aβ40. The protein was modeled with Charmm36m force field parameters and was solvated with TIP3P water molecules. To facilitate exhaustive sampling of Aβ40 in ATP medium, replica exchange MD (REMD) simulation was performed with a total 64 replicas in a temperature range of 290–460 K (see Method). A comparison of 2D-free energy map (**Figure 4A, B**) along $R_g$ and the total number of inter-residue contacts of Aβ40 in aqueous media (later referred as water) containing no ATP and that in presence of 0.5 M aqueous ATP (later referred as ATP) indicates a clear shift of relative population of Aβ40 conformational sub-ensemble toward a basin with higher $R_g$ and lower number of contacts in presence of ATP, suggesting unfolding of the protein chain by ATP. To verify that the effect of ATP on conformational landscape is not an artifact of difference in sampling method (long conventional MD in absence of ATP versus REMD in presence of ATP), we repeated the conformational

sampling in absence of ATP via employing REMD, augmented by adaptive sampling (*Figure 4—figure supplement 1*). We find that the free energy map remains qualitatively similar (*Figure 4A* and *Figure 4—figure supplement 1*) irrespective the sampling technique. Comparison of 2D-free energy map obtained from REMD simulation in absence of ATP (*Figure 4—figure supplement 1*) with the one obtained in presence of ATP (*Figure 4B*) also indicates ATP-driven protein chain elongation. Representative snapshot of the conformation of Aβ40 in aqueous ATP media reveals significantly enhanced disorder in protein (*Figure 4C*). Aβ40 is also very prone to form β-sheet in aqueous medium and thus gets associated in extensive protein aggregation. The central hydrophobic core (residue 17–21) (CHC) (*Jana et al., 2016*) (shown in purple color in *Figure 1*) participates in the β-sheet formation in water and acts as the nucleating core during the pathogenesis relevant to Alzheimer's disease. Interestingly, as evident from the snapshots and from the relative comparison of ensemble-averaged residue-wise β-sheet propensity (*Figure 4D*), ATP reduces the content of β-sheet in CHC region and its partner β-fragment residue 30–35 (SHR, the secondary hydrophobic region). The decrease in the β-sheet signature in aqueous ATP solution is also apparent in the inter-residue contact map (*Figure 4E, F*).

The presence of D23–K28 salt bridge (*Sridharan et al., 2019*; *Tarus et al., 2006*; *Sgourakis et al., 2007*) has been previously reported in the Aβ40 monomer present within aqueous medium. This salt bridge interaction subsequently gives rise to a structural motif which is sufficiently potent for nucleation. D23–K28 salt bridge interaction in Aβ40 has remained crucial in the context of fibrillogenic activities conducted by the pathogenic Aβ40. The salt bridge interaction of D23–K28 stabilizes the β-turn (*Larini and Shea, 2012*) in the region of V24–N27 which consequently favors the hydrophobic contacts between the CHC and the C-terminal part of Aβ40. The structural bend in the zone of residue 23–28 through salt bridge interaction helps the CHC (residues 17–21) and SHR fragments (residues 30–35) to adopt β-sheet formation which gets further stabilized with L17–I32 hydrophobic interaction (*Sridharan et al., 2019*; *Tarus et al., 2006*; *Sgourakis et al., 2007*). All these intramolecular interactions (salt bridge interaction: D23–K28, β-turn: V24–N27, hydrophobic contact: L17–I32) within Aβ40 protein in water, effectively result in the adoption of the β-structure which might be critically responsible for its higher propensity to form pathological amyloid fibrils (*Sridharan et al., 2019*; *Tarus et al., 2006*).

The present simulation trajectory of monomeric Aβ40 in 50 mM aqueous NaCl solution (water) traced all these aforementioned important fibrillogenic interactions. In particular in water the conformational ensemble of Aβ40 adopted D23–K28 salt bridge, V24–N27 β-turn, and L17–I32 hydrophobic interaction (*Figure 5A–F*), as evident from close pairwise distance of separation (*Figure 5—figure supplement 1A–C*). These result in a constrained β-hairpin structure of Aβ40 in water which is capable of inducing aggregation. In presence of ATP (0.5 M ATP in 50 mM NaCl solution), contacts present in these motifs get significantly disfavored (*Figure 5A–F* and *Figure 5—figure supplement 1A–C*), thereby substantially reducing the possibility of aggregation.

Interestingly, ATP's direct interaction (*Figure 5—figure supplement 2*) with the protein molecule allows the disruption of all these pathogenic molecular interactions. *Figure 5G* represents the ATP's region specific (PG, sugar, and base) interaction with the protein Aβ40. The estimation of Coulombic and van der waal interaction energy indicates that unlike Trp-cage (*Figure 3F*), Coulombic interaction of ATP with protein predominates in Aβ40 (*Figure 5H*), partly due to higher proportion of charge in this IDP (see the pie chart for Aβ40 in *Figure 1*). This result signifies that ATP works in the protein-specific manner.

An analysis of solvation-free energy of the protein (see method) in absence and in presence of ATP predicted that the presence of ATP in aqueous solution significantly decreases the free energy required to solvate the IDP (*Figure 5I*), hinting at the ATP's possible role in solubilizing the IDP. This was also evident in the ATP-induced increase in SASA of the protein (*Figure 5—figure supplement 3*).

## Experimental investigation of ATP's effect on aggregation of nucleating core of Aβ40

The aforementioned computer simulation predicts that ATP has the potential to decrease the β-sheet content within the nucleating core of monomeric Aβ40. This finding prompted us to conduct wet-lab experiments to investigate whether ATP plays a role in solubilizing the aggregation-prone Aβ content.

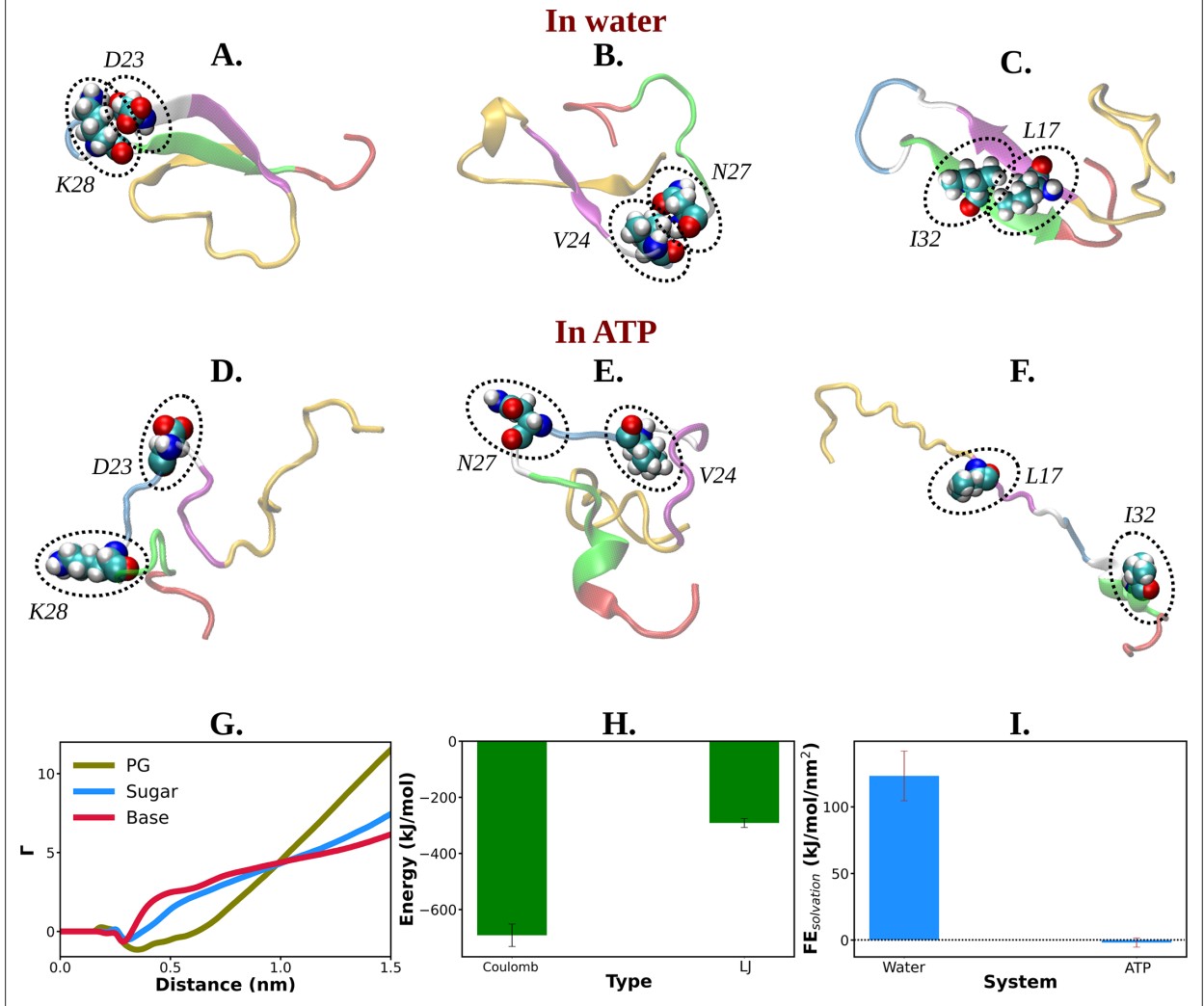

**Figure 5.** Mechanistic insight into ATP's role in the conformational modulation of abeta40. (**A**), (**B**), and (**C**) show the representative snapshots of different pairs of interacting residues namely, D23–K28, V24–N27, and L17–I32, respectively, compared for salt water. The similar set of interactions are being represented in (**D**–**F**) for adenosine triphosphate (ATP) solution containing salt. (**G**) Preferential interaction coefficient (Γ) of different parts of ATP (PG, sugar, and base) with protein is being represented with respect to solvent water. (**H**) The combined coulombic and LJ interaction energies imparted by all the three parts of ATP with Aβ40 are shown. (**I**) The free energy of solvation (calculated by the gromacs module of 'gmx sasa') of Aβ40 protein in absence and in presence of ATP is shown in a bar plot diagram. The vertical lines over the bars show the error bars.

The online version of this article includes the following figure supplement(s) for figure 5:

**Figure supplement 1.** The probability distribution of the distance between the residue pair of D23–K28 (**A**), V24–N27 (**B**), and L17–I32 (**C**) are shown for the protein Aβ40 both in 50 mM NaCl salt solution and 0.5 M aqueous adenosine triphosphate (ATP) solution containing 50 mM NaCl salt.

**Figure supplement 2.** Figure shows a simulation snapshot representing adenosine triphosphate (ATP's) interaction with the Aβ40 protein.

**Figure supplement 3.** The comparison of the solvent-accessible surface area of the protein Aβ40 has been shown for 50 mM NaCl salt solution (blue) and 0.5 M adenosine triphosphate (ATP) in 50 mM NaCl salt (red).

To explore experimental evidence of this influence on the dissolution of misfolded protein, we opted to focus on a shorter subset of the sequence rather than the entire sequence of the intrinsically disordered Aβ (1–40) amyloid. Consequently, we selected a short peptide stretch containing the important nucleating core of Aβ (1–40) from the 16th to the 22nd residue [Ac-KLVFFAE-NH₂, Ac-KE] involving the fibrillogenic CHC region (*Figure 6A*; *Mehta et al., 2008*), which had been suggested by aforementioned computer simulation to impart crucial contribution in formation of β-sheet conformation. This nucleating core is potentially known to act as the intermolecular glue during aggregation. Lysine at the N-terminal and glutamic acid at the C-terminal facilitated an antiparallel arrangement

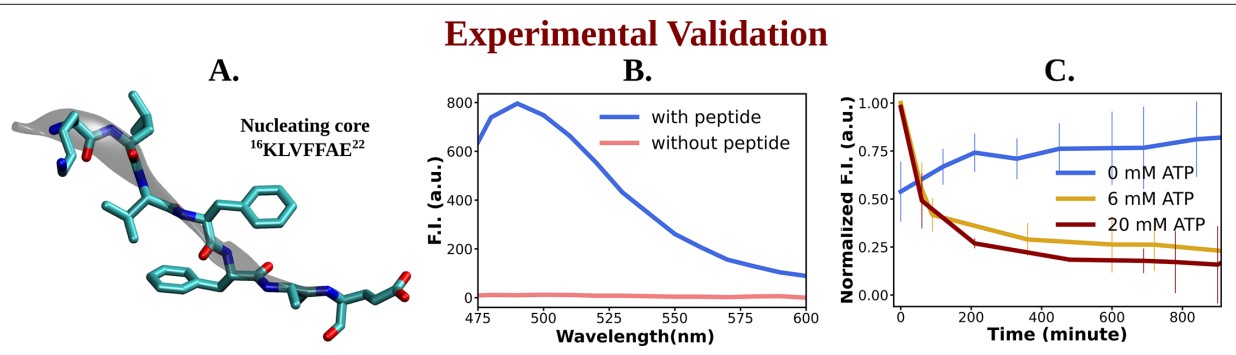

**Figure 6.** Experimental study of ATP's influence on the aggregation of the nucleating core of Aβ40. (**A**) shows the representative snapshot of the peptide belonging to the nucleating core (16th to 22nd residue [Ac-KLVFFAE-NH₂, Ac-KE]) of the Aβ40 protein, which is utilized for the experimental measurements. (**B**) represents emission spectra of Thioflavin T (ThT) in the presence (blue) and absence (pink) of peptide assembly in 10 mM HEPES (4-(2-Hydroxyethyl)piperazine-1-ethane-sulfonic acid) buffer pH 7.2. Excitation wavelength ($\lambda_{ex}$) = 440 nm (final concentration [Ac-KE] = 200 μM, [ThT] = 30 μM). (**C**) shows a comparative plot of ThT (30 μM) assay of Ac-KE (300 μM) assembly with time in 10 mM pH 7.2 HEPES buffer with 0 mM (blue curve), 6 mM (yellow curve), and 20 mM (dark red curve) of adenosine triphosphate (ATP). The vertical lines over the bars show the error bars. The experiments were replicated on two independent samples.

The online version of this article includes the following figure supplement(s) for figure 6:

**Figure supplement 1.** Circular dichroism (CD) spectrum of aged (11–15 days) assembly of Ac-KE.

during assembly formation, contributing to colloidal stability (*Hsieh et al., 2017*). To initiate the assembly of Ac-KE, the synthesized peptide was dissolved in a solution consisting of 40% acetonitrile–water containing 0.1% trifluoroacetic acid (TFA, pH 2). Following an incubation period of approximately 11–15 days, the assemblies exhibited a characteristic β-sheet structure, as revealed by circular dichroism (CD), resembling that of Aβ amyloid (*Figure 6—figure supplement 1*; *Harada and Kuroda, 2011*).

The impact of ATP on amyloid fibrils was investigated using the ThT assay, a well-established method for probing amyloid assembly (*Figure 6B*). In presence of Ac-KE assembly, ThT demonstrated an intense emission at 480 nm which suggested the presence of preformed amyloid aggregation (*Figure 6B*; *Sulatskaya et al., 2011*; *Biancalana et al., 2009*). Various Ac-KE:ATP ratios, ranging from 1:0 to 1:66.7, were tested in HEPES buffer (pH 7.2, 10 mM) at room temperature to assess the influence of ATP on preformed amyloid assemblies (Ac-KE). Interestingly, a progressive reduction in ThT intensity at 480 nm in the presence of ATP (*Figure 6C*) was observed over time, compared to the control system, which contained only the peptide assembly (Ac-KE) in a similar environment (*Figure 6C*). This decrease of ThT intensity in aqueous ATP solution reported the dissolution of the peptide assembly and disruption of the binding sites. Subsequently, morphological changes were examined using transmission electron microscopy (TEM) by casting the incubated samples on the TEM grid (see experimental measurement subsection in Method section). We expected that the investigation from electron microscopy would help in witnessing the visual transformation of fibrillar morphology in the presence of ATP. As a control experiment, the same concentration of Ac-KE assembly was also incubated without ATP in the similar environment for 18 hr. Notably, micrographs recorded at different time frame supported the gradual dissolution of the peptide assembly when samples were incubated with ATP within 18 hr, while the control systems containing only Ac-KE did not demonstrate any noticeable alteration in their fibrillar morphologies (*Figure 7A–C*), library of TEM micrographs: (*Figure 7—figure supplements 1–3*).

## Molecular basis of ATP's solubilizing role of Aβ40 aggregates

The experimental findings indicating ATP's potential role in solubilizing the fibrils of the Aβ40 nucleating core motivated us to explore the molecular basis through the early step of oligomerization. To investigate this, we conducted computational simulations of the dimerization process of Aβ40 in aqueous ATP (0.5 M) media containing 50 mM NaCl and compared it with the same aqueous condition without ATP. Initially, three configurations were considered by placing two copies of the protein

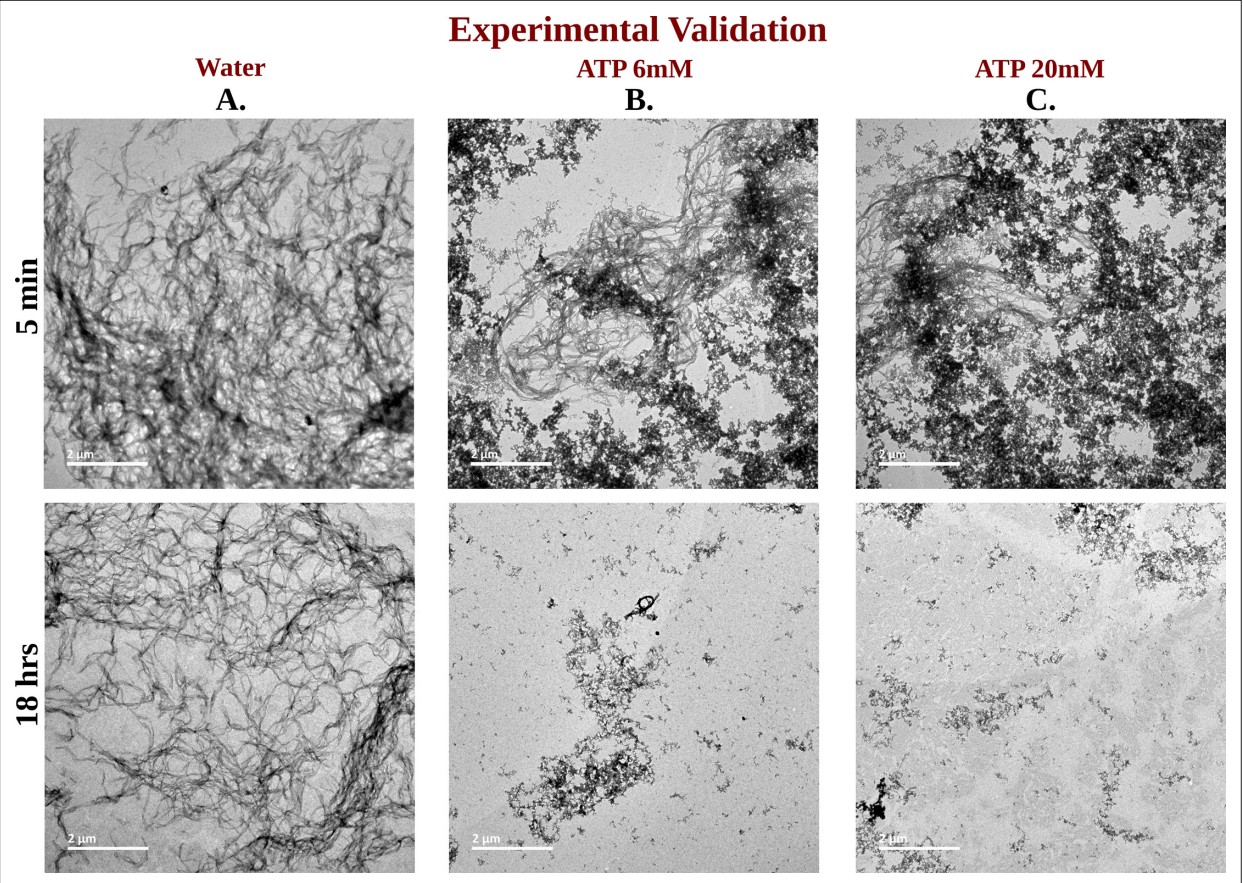

**Figure 7.** TEM Micrographs of Ac-KE Assemblies: Effects of ATP Concentration Over Time. Library of TEM micrographs of Ac-KE (300 µM) assemblies in 10 mM pH 7.2 HEPES buffer at 5 min (up) and after 18 hr (down) of incubation, in presence of (**A**) 0 mM adenosine triphosphate (ATP), (**B**) 6 mM ATP, and (**C**) 20 mM ATP are being represented.

The online version of this article includes the following figure supplement(s) for figure 7:

**Figure supplement 1.** Library of TEM micrographs of 300 µM Ac-KE in 0 mM adenosine triphosphate (ATP) at 5 min (**A–D**) and 18 hr (**E–H**).

**Figure supplement 2.** Library of TEM micrographs of 300 µM Ac-KE in 6 mM adenosine triphosphate (ATP) at 5 min (**A–D**) and 18 hr (**E–H**).

**Figure supplement 3.** Library of TEM micrographs of 300 µM Ac-KE in 20 mM adenosine triphosphate (ATP) at 5 min (**A–D**) and 18 hr (**E–H**).

at a distance (see methods), and unrestrained MD simulations were performed under both conditions, with and without ATP.

The time profiles of the distance between inter-protein contact pairs (involving CHC–CHC, *Figure 8A*, and involving CHC–SHR, *Figure 8B*) revealed that, within the simulation time frame, the pair of Aβ40 approached each other and formed a dimer. However, the propensity for Aβ40 dimerization was significantly diminished in aqueous ATP media. The residue-wise inter-protein contact map of Aβ40 in the absence and presence of ATP molecules (*Figure 8C, D*) further illustrated the disappearance of multiple contact densities in the presence of ATP. Contacts such as CHC–CHC, CHC–SHR, and SHR–SHR (*Fatafta et al., 2021*), which were highly likely to form between protein pairs in aqueous saline medium, did not appear at all in the presence of ATP (*Figure 8D*).

The nucleating core (involving the CHC region) of Aβ40, the same motif used in the previously described wet-lab experiment, played a crucial role as an intermolecular glue. Its active involvement in strong interactions with neighboring Aβ40 protein chains hindered the onset of protein aggregation via the first step of dimerization.

The direct interaction of ATP with Aβ40, playing a pivotal role in attenuating protein–protein interactions, is evident from the preferential interaction coefficient ($\Gamma$) of ATP (*Figure 8E*) with different parts of Aβ40 (NTR, CHC, TR, SHR, and CTR). Notably, ATP exhibits the strongest interactions with CHC and SHR (followed by NTR, TR, and CTR). These regions, as highlighted in the inter-residue

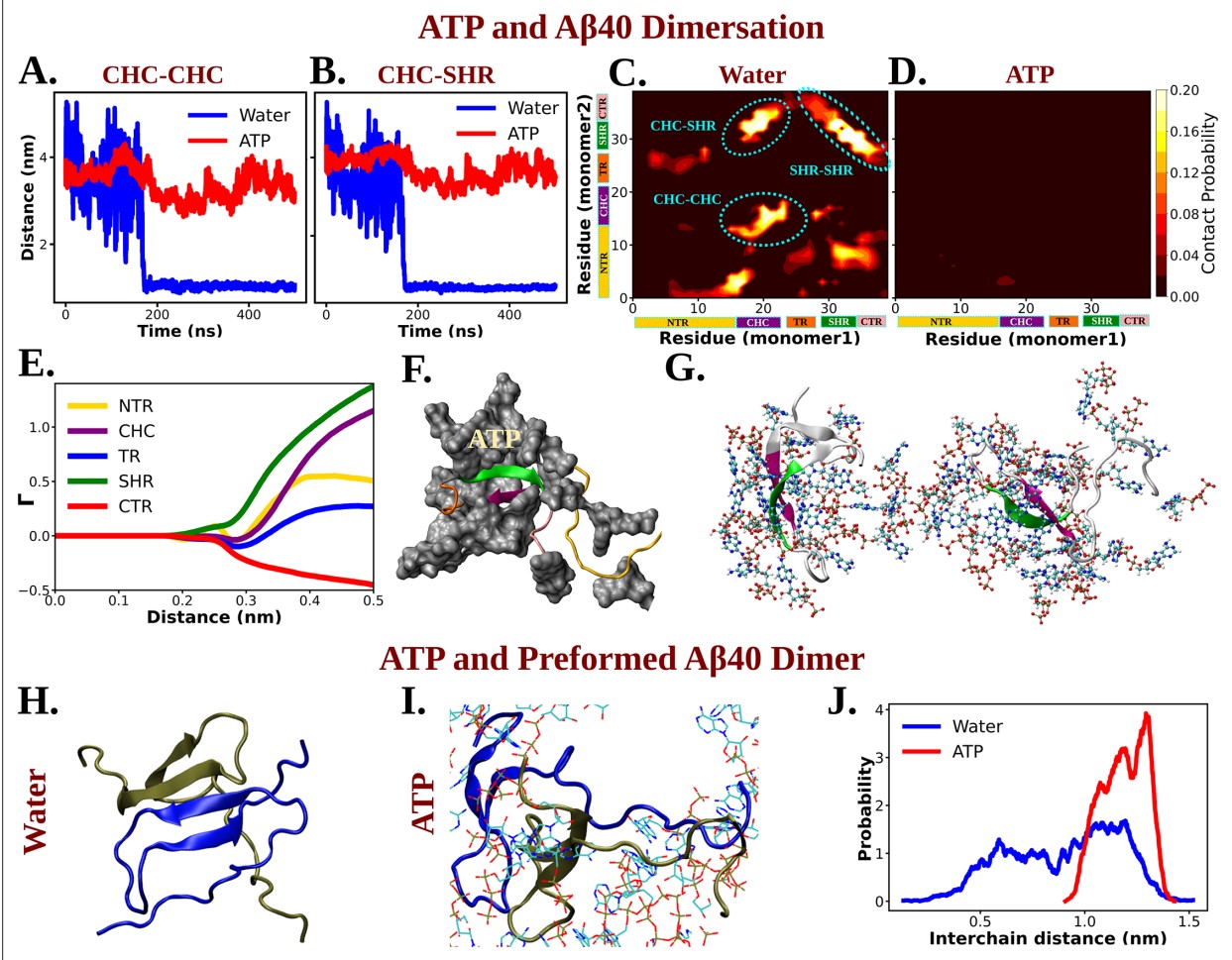

**Figure 8.** Molecular Basis of ATP's Role in Solubilizing Aβ40 Aggregates. (**A**) and (**B**) show the time profile of distance between the two actively interacting regions of two protein chains namely CHC–CHC and CHC–SHR, respectively, both in neat water and in presence of adenosine triphosphate (ATP) cosolute (0.5 M ATP in 50 mM NaCl solution).(**C**) and (**D**) represent residue-wise inter-protein contact map of Aβ40 in water and in 0.5 M ATP solution, respectively. The contacts (CHC–CHC, CHC–SHR, and SHR–SHR) found in neat water are highlighted. (**E**) The preferential interaction coefficient (Γ) of ATP with each different part of Aβ40 protein (NTR, CHC, TR, SHR, and CTR) is being shown. (**F**) Interaction of Aβ40 protein chain with ATP cosolute. ATP molecules are being shown in vdw representation. (**G**) The interacting ATP molecules crowd around the two Aβ40 protein chains are being shown. (**H**) and (**I**) show the consequence of Aβ40 dimer in neat water and in presence of ATP, respectively. The corresponding simulation snapshots are being shown for simulation starting with Aβ40 dimer in water and in 0.5 M ATP in 50 mM NaCl solution. (**J**) represents the probability distribution of distance between the two protein copies of the preformed Aβ40 dimer in absence (50 mM NaCl solution) and presence of ATP (0.5 M ATP in 50 mM NaCl solution).

The online version of this article includes the following figure supplement(s) for figure 8:

**Figure supplement 1.** Residue-level details of ATP's interaction with Abeta40.

contact map in the absence of ATP (*Figure 8C*), predominantly participate in inter-protein interactions during Aβ40 dimerization (refer to *Figure 8F* for a snapshot). A representative snapshot (*Figure 8G*) illustrates the effective crowding of ATP molecules around Aβ40, preventing protein copies from engaging in intermolecular interactions.

The preferential interaction of ATP can also help to lower the probability of intermolecular steric zipper-type interaction (M35–M35) (*Zheng et al., 2007*) which plays a crucial role in aggregation. The inter-chain M35–M35 involving steric zipper interaction is important in forming amyloid fibrils by stabilizing sheet-to-sheet packing with the non-polar zipper. ATP can disrupt the steric zipper interaction through its direct interaction (*Figure 8—figure supplement 1A*) with M35 residue Aβ40 as ATP makes it less available for intermolecular interaction through crowding (*Figure 8—figure supplement 1B*).

## ATP's solubilizing ability of preformed oligomers

ATP not only prevents protein aggregation but can also dissolves pre-existing protein droplets, effectively maintaining proteostasis within cells. To explore ATP's action on a preformed dimer, simulations were initiated with the already-formed protein dimer conformation and simulated its fate in a 50 mM NaCl solution, both with and without the presence of 0.5 M ATP in the same saline medium (*Figure 8H–J*). Interestingly, ATP was found to disrupt the dimer structure (*Figure 8I*), while the structure remained stable in the absence of ATP (*Figure 8H*). The probability distribution (*Figure 8J*) illustrates that, in the presence of ATP, the two protein chains, initially part of the dimer, become prone to be moved away from each other. This demonstrates that, in addition to inhibiting the formation of new protein aggregates, ATP is potent enough to disassemble existing protein droplets, maintaining proper cellular homeostasis. In summary, ATP has been observed to prevent protein condensation, dissolve previously formed condensates, and thereby enhance the solubility of biomolecules within aqueous cellular environments for proper cellular functions.

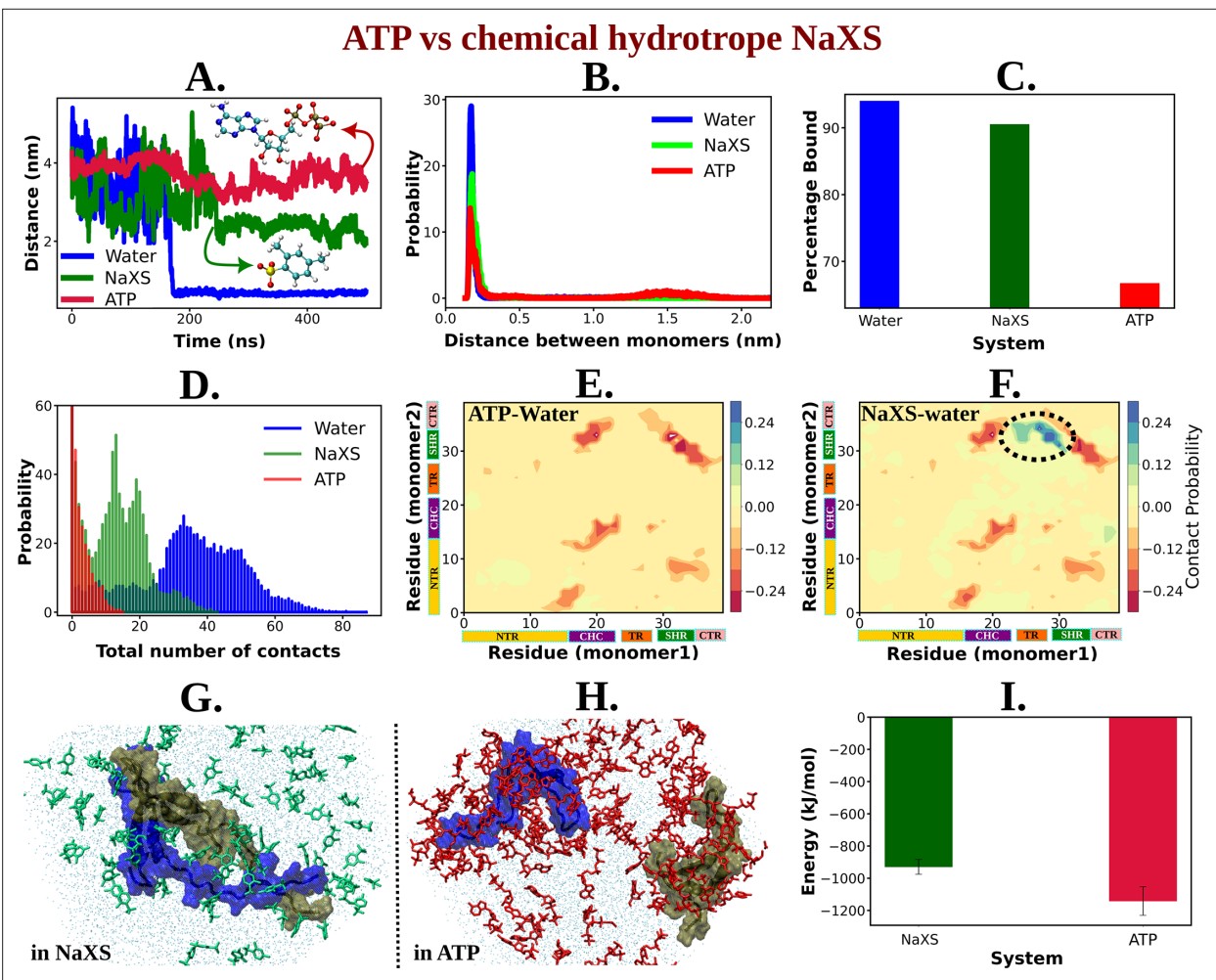

**Figure 9.** ATP's Efficiency as an Aggregate Solubilizer. (**A**) shows the time profile of distance between the two protein chains (CHC–CHC) in neat water (blue curve), in 0.5 M NaXS (green curve) and adenosine triphosphate (ATP) (red curve) in 50 mM NaCl solution. (**B**) represents the probability distribution of the distance between the protein chains in each of the three (above mentioned) cases. (**C**) shows the percentage of bound of the proteins (for all the three systems) in a bar plot representation. (**D**) depicts the total number of intermolecular contacts of the protein monomers in each of the three solutions. (**E**) and (**F**) represent the difference of residue-wise inter-protein contact map of Aβ40 in 0.5 M ATP and 0.5 M NaXS solution, respectively, from that of the neat water system. (**G**) and (**H**) show the representative snapshots captured during the Aβ40 dimerization simulation in 0.5 M NaXS and 0.5 M ATP solution, respectively. *Figure 1* represents the interaction energy (Coulombic interaction) between the NaXS (green bar) and ATP (red bar) molecules with the protein molecules in a bar plot representation. The vertical lines (black colored) show the error bars in the estimation.

## Is ATP special in its role as an aggregate solubilizer?

A crucial question rises: Is ATP special in its role as an aggregate solubilizer? Toward this end, to investigate ATP's relative efficiency in inducing the solubility of hydrophobic entities (biological macromolecules) within cells, we compared it to a conventional chemical hydrotrope sodium xylene sulfonate (NaXS), routinely used in various industrial applications to solubilize sparingly soluble compounds in water. Simulations were conducted with two copies of Aβ40 proteins in a 0.5 M NaXS solution, similar to those performed for 0.5 and 0 M ATP scenarios (discussed earlier; see methods for details).

*Figure 9A* illustrates the time profile of the distance between two protein monomers in 0.5 M NaXS solution compared with that in ATP solution (0.5 M) and neat water. Interestingly, ATP prevents the protein chains from coming closer to each other, while in NaXS solution, the proteins exhibit comparatively closer proximity. This trend is further emphasized by the probability distribution of the inter-protein distance (*Figure 9B*), which shows a gradual decrease in the distance between protein chains from neat water to 0.5 M NaXS solution and finally to 0.5 M ATP. This suggests that, compared to NaXS, ATP is more efficient in enhancing solubility in aqueous medium by preventing the formation of aggregates.

For a quantitative assessment of ATP's efficiency over NaXS, we calculated the bound percentage of protein monomers in each of the three cases (*Figure 9C*). NaXS in water reduces the percentage of bound slightly, indicating a decrease in protein aggregation. However, ATP outperforms NaXS by significantly reducing the propensity of Aβ40 dimerization.

The potency of ATP in inhibiting protein aggregation and maintaining solubility in an aqueous environment, compared to NaXS, is further evident from the estimation of the total number of inter-protein contacts (*Figure 9D*). As we move from neat water to 0.5 M NaXS solution and finally to 0.5 M ATP solution, the number of intermolecular contacts between the two protein chains gradually decreases, highlighting ATP's efficiency over NaXS.

The contact map (residue-wise inter protein contact map) difference (contact map of ATP/NaXS + water from that in neat water only) reveals that in the presence of ATP, there is a significant decrease in the probability of inter-protein contacts (*Figure 9E*), a crucial factor in protein aggregation (*Figure 8C*). In contrast, in the presence of NaXS (*Figure 9F*), along with a decrease in some contact probabilities, there is an increase in inter-protein contacts, especially involving the CHC and TR regions of one chain with the SHR region of the other protein.

In nutshell, ATP acts more efficiently compared to NaXS by effectively inhibiting inter-protein interactions, preventing aggregation, and maintaining protein solubility in an aqueous environment (*Figure 9G, H*). To elucidate the factors contributing to ATP's superiority over NaXS, we examined the interaction energy (Coulombic interaction) of both ATP and NaXS with the protein molecules (*Figure 9I*). The analysis revealed that, in comparison to NaXS, ATP exhibits a stronger interaction with the proteins, resulting in higher interaction energy. This heightened interaction energy plays a crucial role in preventing protein–protein interactions, ultimately leading to the effective inhibition of aggregation.

Comparing the effects of ATP with other nucleotides such as ADP and GTP, we emphasize that previous studies have demonstrated GTP can dissolve protein droplets (such as FUS) with efficiency comparable to ATP. However, in vivo, the concentration of GTP is significantly lower than that of ATP, resulting in negligible impact on the solubilization of liquid compartments. In contrast, ADP and AMP show much lower efficiency in dissolving protein condensates, indicating the critical role of the triphosphate moiety in protein condensate dissolution. Furthermore, only TP-Mg exhibited a negligible effect on protein droplet dissolution, suggesting that the charge density in the ionic ATP side chain alone is insufficient for this process. These findings underscore ATP's superior efficacy as a protein aggregate solubilizer, attributed to its specific chemical structure rather than merely its amphiphilicity.

## Discussion

While previous efforts have explored ATP-driven protein disaggregation, its impact on protein conformational plasticity, crucial for self-assembly, is understudied. Our computational investigation prioritizes the examination of ATP's influence on protein conformational thermodynamics and correlates it with preventing protein aggregation. Regardless of biomolecular structure (globular or intrinsically

disordered), ATP is found to induce protein chain extension, disrupting specific secondary structures and promoting flexibility for solvation in water. We demonstrate ATP's direct enhancement of protein aqueous solubility through calculations of solvation-free energy and SASA. The substantial increase in accessible surface area and more negative solvation-free energy in aqueous ATP solution compared to ATP-free solution highlights ATP's pivotal role in preventing protein aggregation by increasing aqueous solubility through protein chain extension.

We followed up the simulation's prediction on ATP's role on protein conformation via a set of proof-of-concept experiments, in which we investigated ATP's impact on the fibrillogenic nucleating core of Aβ40 (residues 16th to 22nd). ThT assays revealed a monotonically decreasing intensity over time with increasing ATP concentrations (0–20 mM), indicating the dissolution of Aβ40 peptide assemblies. TEM confirmed the dissolution of peptide assemblies within 18 hr in ATP-treated samples, contrasting with control systems containing only Ac-KE, which showed no significant alteration in fibrillar morphologies. This experimental evidence, coupled with insights into ATP's role in enhancing aqueous solubility, necessitates an understanding of its molecular inhibitory mechanisms at the oligomerization level (*Kurisaki and Tanaka, 2019*).

Computational studies focused on the primitive aggregation stage, specifically Aβ40 dimerization, revealed that ATP disrupts dimer formation compared to the rapid dimerization observed in water. ATP's ability to guide conformational changes, particularly by disfavoring β-sheet adoption between the CHC and SHR fragments, efficiently inhibits Aβ40 dimerization. Direct interactions between ATP molecules and the protein were identified as crucial, preventing pathological interactions (CHC–CHC, CHC–SHR, and SHR–SHR) associated with the β-sheet motif. Additionally, ATP not only inhibits aggregation but also disintegrates previously formed dimers, highlighting its capability to dissolve existing protein aggregates. Overall, ATP's multifaceted action, from inhibiting dimerization to disrupting existing dimers, underscores its potential in preventing pathological protein aggregation.

In conclusion, beyond its role in various energetic modulations, ATP contributes significantly and independently of energy expenditure to the proper functioning of proteins in crowded cellular environments with high salt concentrations, mitigating biomolecular instabilities linked to diseases involving pathological protein aggregation. Our study demonstrates that ATP's micromanagement starts at the protein monomeric level, orchestrating conformational modifications to ensure adequate aqueous solubility. By perturbing intramolecular contacts, ATP induces structural flexibility, facilitating effective water solubilization of protein monomers. Simultaneously, ATP discourages the adoption of fibrillogenic protein conformations prone to aggregation. Recently described as hydrotropic activity, our findings provide a detailed understanding of the biological rationale for the exceptionally high concentration of ATP in cells compared to other nucleotides, as reported by *Patel et al., 2017*. While previous studies highlighted ATP's antiaggregation property, our investigation elucidates its mechanism: ATP's remarkable influence on protein conformational modification helps hydrophobic protein molecules remain favorably soluble in the cellular aqueous medium, thereby reducing their propensity for self-aggregation. Overall, ATP's regulatory role in conformational plasticity emerges as a key factor behind its hydrotropic function in biological systems.

The mechanistic details of ATP's interaction with proteins have been subjected to various interpretations. Few studies proposed a nonspecific ATP–protein interaction (*Nishizawa et al., 2021*; *Zalar et al., 2023*), contrasting with the some reports explaining ATP's preferential binding to conserved residues (Arg and Lys or Tyr and Ser) (*Kim et al., 2021*; *Song, 2021*; *Coskuner and Murray, 2014*). Some studies highlighted ATP's role as a solvation mediator with the capabilities of ATP in hydrophobic, π–π, π–cation, and electrostatic interactions with proteins (*Sridharan et al., 2019*; *Pal and Paul, 2020*; *Sarkar and Mondal, 2021*; *Dec et al., 2023*). In our investigation, we found that ATP interacts directly with Trp-cage and Aβ40 proteins in a protein-specific manner, utilizing its chemically distinct parts—negatively charged PG group, hydrophilic sugar moiety, and hydrophobic aromatic base—to engage in electrostatic, H-bond-like, and hydrophobic interactions with protein residues. The nature of ATP's interactions varies significantly based on the amino-acid composition of the protein. For the hydrophobic Trp-cage, ATP favors van der Waals' interactions over electrostatic ones, while for the charged Aβ40, Coulombic interactions dominate. This protein-dependent specificity enhances ATP's efficiency in cellular processes, aligning with recent proteome-wide (*Sridharan et al., 2019*) studies showing diverse effects of ATP on protein thermal stability and solubility. Notably, IDPs experience enhanced solubility with ATP, while some proteins exhibit decreased solubility.

In our literature survey of ATP's concentration-dependent actions, as detailed in the Introduction section, we observed a dual role where ATP induces protein liquid–liquid phase separation at lower concentrations and promotes protein disaggregation at higher concentrations (*Song, 2021*; *Ren et al., 2022*; *Liu and Wang, 2023*). These versatile functions emphasize ATP's pivotal role in maintaining a delicate balance between protein stability (at low ATP concentrations) and solubility (at high ATP concentrations) for effective proteostasis within cells. Notably, ATP-mediated stabilization primarily targets soluble proteins, particularly those with ATP-binding motifs, while ATP-driven biomolecular solubilization is observed for insoluble proteins, typically lacking ATP-binding motifs. The question arises: how does ATP selectively stabilize or destabilize proteins? Recent proteome-based (*Sridharan et al., 2019*) investigations align with previous study (*Saraste et al., 1990*) indicating that ATP predominantly stabilizes soluble biomolecules, especially those featuring the P-loop motif (GK[X]nS/T or G[X]nK[X]nK). Our sequence-based analysis of reported proteins, including ubiquitin, malate dehydrogenase, and TDP-43 RRM domain, reveals conservation of the P-loop motif in their respective ATP recognition site (see *Figure 1—source data 1*). Conversely, proteins like Trp-cage and Aβ40, undergoing ATP-driven solubilization at the expense of conformational stability, lack the P-loop motif (see *Figure 1—source data 1*). This suggests that ATP deciphers information from the primary structure (protein sequence) to govern secondary and tertiary structures, selectively stabilizing ATP-binding proteins and inducing solubilization in the absence of ATP-binding motifs during protein unfolding.

In summary, while previous reports emphasize ATP's role in inhibiting protein aggregation, our work connects these findings by highlighting ATP's influence starting at the monomeric level, thereby preventing proteins from becoming aggregation prone. The distinctive protein-dependent and region-specific interactions between ATP and protein molecules play a pivotal role in regulating protein structure. This interaction mechanism leads to an exceptionally efficient inhibition of protein aggregation compared to conventional chemical hydrotropes. ATP orchestrates a two-stage hydrotropic action, from monomer destabilization to early-stage aggregation prevention. The necessity for higher cellular concentrations of ATP underscores its significance, potentially explaining the frequent onset of protein aggregation-related diseases with aging, attributed to ATP deficiency. The collaborative efforts of simulation and experimentation in this study suggest potential implications and therapeutic interventions using ATP for future treatments of various neurodegenerative diseases.

## Materials and methods
### Simulation model and methods

In this work, we have studied the effect of ATP on the conformational plasticity of two different types of proteins belonging from the contrasting protein spectrum: (1) folded globular protein, Trp-cage which stands as one of the very suitable prototypical computational mini-protein models with less charge content and (2) the IDP, Aβ40 containing higher proportion charged residues (in line with the typical nature of IDP), which is popularly known for its characteristic high aggregation propensity (*Figure 1*). Here, we have individually investigated the conformational dynamics of both the proteins in absence and presence of ATP-Mg$^{2+}$. The folded protein Trp-cage, PDB: 1L2Y (both the N- and C-terminal of the protein was capped with acetyl group and methyl amide, respectively) was studied in neat water and in 0.1 and 0.5 M aqueous ATP solution (protein/ATP stoichiometry of $0.02 \times 10^3$). Further equivalent number of Mg$^{2+}$ ions was incorporated into the simulation box. The system was charge neutralized with an equivalent number of chloride ions. The protein was modeled with a99SB-*disp* force field parameters and solvated with TIP4P-*disp* water molecules (*Robustelli et al., 2018*). The ion parameters were taken from a99SB-*disp* force field. During simulation with ATP-Mg$^{2+}$ (0.1 and 0.5 M), ATP molecules were modeled with Amber force field parameters. For Trp-cage simulations in neat water and 0.5 M ATP-Mg$^{2+}$ aqueous solution the box volume was $4.4 \times 4.4 \times 3.1$ nm$^3$ and for 0.1 M ATP-Mg$^{2+}$ simulation with Trp-cage the box dimension was kept fixed as $8.2 \times 8.2 \times 5.7$ nm$^3$.

First each system was energy minimized followed by two consecutive steps of equilibration: (1) NVT equilibration for 1 ns at an equilibrium temperature of 300 K utilizing the Nosé–Hoover thermostat (*Nosé, 1984*; *Hoover, 1985*) (1 ps time constant) and then (2) equilibration in NPT ensemble for 2 ns at 300 K temperature and 1 bar pressure maintained using the Nosé–Hoover thermostat (time constant of 1 ps) and Berendsen barostat (*Berendsen et al., 1987*) (time constant of 1.0 ps),

respectively. Further, the system was again equilibrated in the NPT ensemble for 10 ns at an equilibrium temperature of 300 K and an equilibrium pressure of 1 bar, maintained by employing the Nosé–Hoover thermostat (time constant of 1 ps) and the Parrinello–Rahman barostat (*Parrinello and Rahman, 1981*) (time constant of 1.0 ps), respectively. The particle-mesh Ewald (*Parrinello and Rahman, 1981*; *Darden et al., 1993*) method with a grid spacing of 0.12 nm was applied for managing long-range electrostatics. For constraining the bonds associated with hydrogen atoms and the bonds and angle of water molecules, the LINCS method (*Hess, 2008*) and SETTLE algorithm (*Hess, 2008*; *Miyamoto and Kollman, 1992*), respectively, were used. To perform all the MD simulations GROMACS (*Abraham et al., 2015*) software of version 2018.6 software was utilized.

To ensure exhaustive exploration of protein conformational landscape conditions, each condition (Trp-cage in neat water and 0.1 and 0.5 M ATP-Mg$^{2+}$ aqueous solution of Trp-cage) was simulated via REMD simulation (*Sugita and Okamoto, 1999*). For performing REMD simulation the former equilibrated systems were employed. REMD simulation was done in the temperature range of 280–540 K with a total of 54 replicas (for 0.1 M ATP-Mg$^{2+}$ solution the temperature range was 290–460 K with 64 replicas). At first each of the replicas was well equilibrated for 5 ns in the NVT ensemble at the respective replica temperature employing the Nosé–Hoover thermostat (time constant 1 ps). Eventually the production simulations were performed for 400 ns (250 ns for 0.1 M ATP-Mg$^{2+}$ solution of Trp-cage) at each replica with a replica exchange interval of 10 ps. Finally, the trajectory corresponding to the replica temperature of 303 K was considered for subsequent analysis. Further to test the robustness of our estimation across the choice of force field, the similar set of simulations for the Trp-cage protein in neat water and in 0.5 M ATP are performed with Charmm36 force field parameters for protein (*Best et al., 2012*) and ATP (*Hart et al., 2012*) and using Charmm TIP3P water model (*Sarkar and Mondal, 2021*; *MacKerell et al., 1998*). For these simulations in the Charmm36 force field the similar simulation protocol has been utilized.

Apart from folded globular protein, for studying ATP's effect on the conformation plasticity of an IDP we have chosen Aβ40 which is routinely investigated in the context of protein aggregation-related pathogenic conditions. Accordingly, the 30 μs long MD simulation trajectory of monomeric Aβ40 in 50 mM aqueous NaCl media, from D. E. Shaw research was utilized (*Robustelli et al., 2018*). To realize the impact of ATP on the conformational dynamics of Aβ40, we simulated the conformational repotiare of monomeric form of Aβ40 (initial structure same as D. E. Shaw research group) in 0.5 M (ATP/protein stoichiometry $0.1 \times 10^3$) aqueous ATP solution containing 50 mM NaCl salt. At first the protein was incorporated in the simulation box of dimension $8.2 \times 8.2 \times 5.7$ nm$^3$ followed by addition of ATP molecules to maintain ATP concentration of 0.5 M. Equivalent numbers of Mg$^{2+}$ ions were added. Finally, the system was solvated with water molecules sufficient to fill up the box. The protein was modeled with Charmm36m force field parameters *Robustelli et al., 2018* and for water molecules, Charmm TIP3P water model was employed (*Sarkar and Mondal, 2021*; *MacKerell et al., 1998*). The ion parameters are obtained from the Charmm 36m force field. ATP molecule is modeled with Charmm36 force field parameters (*Hart et al., 2012*). For exhaustive sampling of the protein in ATP medium, REMD simulation was performed with a total 64 replicas in a temperature range of 290–460 K. Each of the replicas was simulated for 400 ns at the replica exchange interval of 10 ps, leading to a cumulative sampling aggregate equivalent to around 26 μs. The REMD conformations corresponding to 303 K were further clustered into 213 clusters via the $R_g$ and total number of inter-residue contacts of the protein chain based on regular space clustering algorithm. Finally, 213 independent MD simulations were performed for 100 ns (each) with the different initial configurations (Aβ monomer dissolved in 0.5 M ATP in 50 mM aqueous NaCl solution) chosen randomly from individual clusters. The concatenated short trajectories were employed for generating 2D energy profiles and for all other calculation 303 K REMD simulation trajectory was employed. The 213 short MD simulation trajectories are used separately for building a Markov state model (MSM) (*Husic and Pande, 2018*) in order to statistically map the complete process of protein conformational change using PyEMMA software (*Husic and Pande, 2018*; *Scherer et al., 2015*). From MSM, the stationary populations of the discrete microstates were calculated and eventually utilized for reweighing the free energy surfaces obtained from these short trajectories. To test the robustness, we have also estimated the 2D-free energy profile of Aβ40 in absence of ATP by performing similar REMD simulation followed by adaptive sampling simulation following the similar protocol described above.

Further to test the correlation of ATP's effect on protein monomeric level with its potency to inhibit aggregation, we have carried out equilibrium dimer simulations with the aggregation-prone amyloidogenic protein aβ40. We have started our simulation from three different conformations involving the protein chains at three different distances 1, 2, and 4 nm. Two replica simulations are performed with each of the configurations for 500 ns following the similar simulation protocol as mentioned above. The similar set of dimer simulations were performed for the two copies of the Aβ40 proteins in the 50 mM aqueous NaCl solution and also 0.5 M aqueous ATP solution containing 50 mM NaCl salt. The same box size was maintained for each of the dimer simulations as 8.2 × 8.2 × 5.7 nm$^3$ (same as the monomeric Aβ40 simulation with ATP-Mg$^{2+}$). Later we have repeated the similar dimer simulations for 0.5 M NaXS in the 50 mM aqueous NaCl solution, following the similar simulation protocol.

To assess the impact of ATP on the pre-existing protein droplets, we have tested ATP's effect on the preformed dimer. We have started our simulation with preformed Aβ40 dimer (three dimer conformations were obtained from the previously described Aβ40 dimerization simulation in 50 mM NaCl salt solution and simulations were carried out corresponding to each of the dimer conformation) dissolved in 0.5 M ATP-Mg$^{2+}$ in 50 mM aqueous NaCl solution. Simulation was performed in the same box size of 8.2 × 8.2 × 5.7 nm$^3$ following the similar simulation protocol. Simulation was continued up to 1 μs.

GROMACS software analysis tools were utilized for computing the radius of gyration ($R_g$), number of hydrogen bonds of the proteins. The fraction of native contacts (for Trp-cage) was estimated using PLUMED software (*Tribello et al., 2014*) with a cutoff of 0.7 nm. For calculating the total number of contacts and residue-wise contact map MDAnalysis (*Gowers et al., 2016*) tool was employed along with python scripting. The secondary structure content of each of the proteins was estimated using the STRIDE program of VMD. The calculation of the SASA was performed with the help of the GROMACS analysis tool (gmx sasa). The 2D-free energy profiles were computed for each of the systems using the PyEMMA tool. The hydrophobicity index of each of the proteins was obtained from the peptide 2.0 web server 'https://www.peptide2.com/N_peptide_hydrophobicity_hydrophilicity.php'. For quantitative estimation of ATP's interaction with the proteins, the Wyman–Tanford preferential interaction coefficient (*Wyman, 1964*; *Tanford, 1969*) ($\Gamma_s$) was calculated (*Sarkar et al., 2022*; *Sarkar et al., 2023a*; *Mondal et al., 2015*) with respect to water.

$$\Gamma_s = \left\langle n_s - \frac{N_s^{tot} - n_s}{N_w^{tot} - n_w} \cdot n_w \right\rangle$$

where $n_s$ is the number of cosolute bound with the protein molecule and $N_s^{tot}$ is the total number of the species present within the system. $n_w$ represents the number of water molecules bound to the surface of the protein and $N_w^{tot}$ is the total number of water of the system.

## Experimental measurements

### Materials
All protected amino acids, activator *N,N'*-diisopropylcarbodiimide (DIC), *N,N*-diisopropylethylamine, TFA, piperidine, uranyl acetate, ThT, and HEPES were purchased from Sigma-Aldrich. Oxyma was purchased from Nova Biochem. All solvents and Fmoc-Rink amide MBHA Resin were purchased from Merck. Milli-Q water was used throughout the experiments.

### Peptide synthesis and characterization
Ac-KLVFFAE-NH$_2$ (Ac-KE) peptide was synthesized by solid-phase peptide synthesizer (CEM Liberty Blue). Fmoc-Rink Amide MBHA resin (loading 0.52 mmol/g) was swollen using dimethylformamide (DMF) for 15 min, followed by Fmoc deprotection with 20% piperidine in DMF. Each Fmoc-amino acid coupling step was performed using DIC and oxyma pure in DMF. Subsequently, acetylation of the N-terminus lysine was done using acetic anhydride in DMF. The resin was washed with dichloromethane after completion of the final coupling and allowed to dry in open air. Peptide was cleaved from the resin using TFA/triethyl silane (5:0.1 vol/vol) solution for 2 hr, followed by precipitation in cold diethyl ether after removing the TFA. The product was centrifuged at 5000 rpm for 15 min at 4°C in

Eppendorf centrifuge 5804 R and further, the pellet was washed three times with cold diethyl ether. Molecular mass was confirmed by Waters Xevo G2-XS QTof.

Ac-KLVFFAE-NH$_2$ (Ac-KE) (C$_{45}$H$_{67}$N$_9$O$_{10}$) ($m/z$) calculated for [M+H$^+$]: 894.50; found: 894.52.

## Peptide assembly

The dried powder of the peptide (Ac-KLVFFAE-NH$_2$) was treated with HFIP to eliminate preformed assembly during precipitation in ether. After that, HFIP was removed through N$_2$ blowing and the peptide film was dissolved in 40% acetonitrile–water containing 0.1% TFA through vortexing and sonication. The homogeneous solution of peptide was kept for 11–15 days at ambient temperature (~2–8°C) to form assembly formation.

## Circular dichroism

A JASCO J-810 circular dichroism spectrometer fitted with a Peltier temperature controller to maintain the temperature at 25°C was used to record the CD spectra. 200 µM of Ac-KE (stock 2.5 mM) aged assemblies was taken in Mili-Q water and the spectra were recorded in a quartz cuvette with a 1 mm path length. Spectrum was recorded throughout the wavelength range from 300 to 180 nm with a scan rate of 100 nm/min and two accumulations.

## ThT assay

The 11–15 days matured assembly Ac-KE peptide (300 µM) was mixed with different concentrations of ATP (6 and 20 mM) in 10 mM of HEPES buffer (pH 7.2) at room temperature. 30 µM of ThT dye was added to the mixture and started the experiment with excitation wavelength ($\lambda_{ex}$) at 440 nm and emission ($\lambda_{em}$) at 480 nm. Kinetics was recorded for 18 hr with 30-min intervals, and 30 s constant shaking was set before taking the data. The gain was fixed to 50 in the microplate reader (BioTek, SYNERGY H1). To check the control, only Ac-KE (300 µM) was taken in a similar environment and data were recorded.

To check ThT as a reporter for amyloid assembly fluorescence spectra in microplate reader was checked for 200 µM of Ac-KE assembly in presence of 30 µM ThT. The excitation wavelength and gain were set at 440 nm and 50, respectively.

## Acknowledgements

We acknowledge support of the Department of Atomic Energy, Government of India, under Project Identification No. RTI 4007. We also acknowledge Core Research grants provided by the Department of Science and Technology (DST) of India (CRG/2023/001426).

# Additional information

### Funding

| Funder | Grant reference number | Author |
| --- | --- | --- |
| Department of Atomic Energy, Government of India | RTI 4007 | Susmita Sarkar<br>Saurabh Gupta<br>Chiranjit Mahato<br>Dibyendu Das<br>Jagannath Mondal |
| Department of Science and Technology, Ministry of Science and Technology, India | CRG/2023/001426 | Jagannath Mondal |

The funders had no role in study design, data collection, and interpretation, or the decision to submit the work for publication.

## Author contributions
Susmita Sarkar, Conceptualization, Data curation, Formal analysis, Methodology, Writing – original draft, Writing – review and editing; Saurabh Gupta, Data curation, Formal analysis, Validation, Methodology; Chiranjit Mahato, Data curation, Formal analysis, Methodology; Dibyendu Das, Formal analysis, Supervision, Validation, Methodology, Project administration, Writing – review and editing; Jagannath Mondal, Conceptualization, Resources, Formal analysis, Supervision, Validation, Writing – original draft, Project administration, Writing – review and editing

## Author ORCIDs
Susmita Sarkar (iD) http://orcid.org/0009-0001-1105-1466
Jagannath Mondal (iD) https://orcid.org/0000-0003-1090-5199

Reviewer #1 (Public review): https://doi.org/10.7554/eLife.99150.3.sa1
Reviewer #3 (Public review): https://doi.org/10.7554/eLife.99150.3.sa2
Author response https://doi.org/10.7554/eLife.99150.3.sa3

# Additional files

## Supplementary files
• MDAR checklist

## Data availability
The source files of the data presented in this article and the key computer simulation trajectories related to the investigation have been uploaded to zenodo (https://zenodo.org/records/13831366) for public access.

The following dataset was generated:

| Author(s) | Year | Dataset title | Dataset URL | Database and Identifier |
|---|---|---|---|---|
| Sarkar S | 2024 | Raw data for "Elucidating ATP's Role as Solubilizer of Biomolecular Aggregate" | https://doi.org/10.5281/zenodo.13831366 | Zenodo, 10.5281/zenodo.13831366 |

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
