## [Editor Report · eLife assessment]

The authors combined molecular dynamics simulations and experiments to study the role of ATP as a hydrotrope of protein aggregates. The topic is of major current interest and thus the study potentially makes an **important** contribution to the community. With the revised version, the level of evidence is considered generally **solid**, although there remains concern regarding the unusually high ATP concentration used in the simulation.

---

## [Referee Report · Reviewer #1 (Public review)]

Summary:

This work combines molecular dynamics (MD) simulations along with experimental elucidation of the efficacy of ATP as biological hydrotrope. While ATP is broadly known as the energy currency, it has also been suggested to modulate the stability of biomolecules and their aggregation propensity. In the computational part of the work, the authors demonstrate that ATP increases the population of the more expanded conformations (higher radius of gyration) in both a soluble folded mini-protein Trp-cage and an intrinsically disordered protein (IDP) Aβ40. Furthermore, ATP is shown to destabilise the pre-formed fibrillar structures using both simulation and experimental data (ThT assay and TEM images). They have also suggested that the biological hydrotrope ATP has significantly higher efficacy as compared to the commonly used chemical hydrotrope sodium xylene sulfonate (NaXS).

Strengths:

This work presents a comprehensive and compelling investigation of the effect of ATP on the conformational population of two types of proteins: globular/folded and IDP. The role of ATP as an "aggregate solubilizer" of pre-formed fibrils has been demonstrated using both simulation and experiments. They also elucidate the mechanism of action of ATP as a multi-purpose solubilizer in a protein-specific manner. Depending on the protein, it can interact through electrostatic interactions (for predominantly charged IDPs like Aβ40), or primarily van der Waals' interactions through (for Trp-Cage).

Weaknesses:

The weaknesses and suggestions mentioned in my first review have been adequately addressed by the authors in the revised version of the manuscript.

---

## [Referee Report · Reviewer #3 (Public review)]

Since its first experimental report in 2017 (Patel et al. Science 2017), there have been several studies on the phenomenon in which ATP functions as a biological hydrotrope of protein aggregates. In this manuscript, by conducting molecular dynamics simulations of three different proteins, Trp-cage, Abeta40 monomer, and Abeta40 dimer at concentrations of ATP (0.1, 0.5 M), which are higher than those at cellular condition (a few mM), Sarkar et al. find that the amphiphilic nature of ATP, arising from its molecular structure consisting of phosphate group (PG), sugar ring, and aromatic base, enables it to interact with proteins in a protein-specific manner and prevents their aggregation and solubilize if they aggregate. The authors also point out that in comparison with NaXS, which is the traditional chemical hydrotrope, ATP is more efficient in solubilizing protein aggregates because of its amphiphilic nature.

Trp-cage, featured with hydrophobic core in its native state, is denatured at high ATP concentration. The authors show that the aromatic base group (purine group) of ATP is responsible for inducing the denaturation of helical motif in the native state.

For Abeta40, which can be classified as an IDP with charged residues, it is shown that ATP disrupts the salt bridge (D23-K28) required for the stability of beta-turn formation.

By showing that ATP can disassemble preformed protein oligomers (Abeta40 dimer), the authors suggest that ATP is "potent enough to disassemble existing protein droplets, maintaining proper cellular homeostasis," and enhancing solubility.

Overall, the message of the paper is clear and straightforward to follow. In addition to the previous studies in the literature on this subject. (J. Am. Chem. Soc. 2021, 143, 31, 11982-11993; J. Phys. Chem. B 2022, 126, 42, 8486-8494; J. Phys. Chem. B 2021, 125, 28, 7717-7731; J. Phys. Chem. B 2020, 124, 1, 210-223), the study, which tested using MD simulations whether ATP is a solubilizer of protein aggregates, deserves some attention from the community and is worth publishing.

Weakness

My only major concern is that the simulations were performed at unusually high ATP concentrations (100 and 500 mM of ATP), whereas the real cellular concentration of ATP is 1-5 mM.

I was wondering if there is any report on a titration curve of protein aggregates against ATP, and what is the transition mid-point of ATP-induced solubility of protein aggregates. For instance, urea or GdmCl have long been known as the non-specific denaturants of proteins, and it has been well experimented that their transition mid-points of protein unfolding are in the range of ~(1 - 6) M depending on the proteins.

The authors responded to my comment on ATP concentration that because of the computational issue in all-atom simulations, they had no option but to employ mM-protein concentrations instead of micromolar concentrations, thus requiring 1000-folds higher ATP concentration, which is at least in accordance with the protein/ATP stoichiometry. However, I believe this is an issue common to all the researchers conducting MD simulations. Even if the system is in the same stoichiometric ratio, it is never clear to me (is it still dilute enough?) whether the mechanism of solubilization of aggregate at 1000 fold higher concentration of ATP remains identical to the actual process.

---

## [Author Response]

The following is the authors’ response to the current reviews.

**Reviewer #1 (Public review):**
Summary:This work combines molecular dynamics (MD) simulations along with experimental elucidation of the efficacy of ATP as biological hydrotrope. While ATP is broadly known as the energy currency, it has also been suggested to modulate the stability of biomolecules and their aggregation propensity. In the computational part of the work, the authors demonstrate that ATP increases the population of the more expanded conformations (higher radius of gyration) in both a soluble folded mini-protein Trp-cage and an intrinsically disordered protein (IDP) Aβ40. Furthermore, ATP is shown to destabilise the pre-formed fibrillar structures using both simulation and experimental data (ThT assay and TEM images). They have also suggested that the biological hydrotrope ATP has significantly higher efficacy as compared to the commonly used chemical hydrotrope sodium xylene sulfonate (NaXS).Strengths:This work presents a comprehensive and compelling investigation of the effect of ATP on the conformational population of two types of proteins: globular/folded and IDP. The role of ATP as an "aggregate solubilizer" of pre-formed fibrils has been demonstrated using both simulation and experiments. They also elucidate the mechanism of action of ATP as a multi-purpose solubilizer in a protein-specific manner. Depending on the protein, it can interact through electrostatic interactions (for predominantly charged IDPs like Aβ40), or primarily van der Waals' interactions through (for Trp-Cage).Weaknesses:The weaknesses and suggestions mentioned in my first review have been adequately addressed by the authors in the revised version of the manuscript.

Thank you very much for your positive feedback and for taking the time to thoroughly review our manuscript. Your thoughtful comments and suggestions have significantly contributed to enhancing the quality of our work.

We sincerely appreciate your time and efforts in helping us refine our research.

**Reviewer #3 (Public review):**
Since its first experimental report in 2017 (Patel et al. Science 2017), there have been several studies on the phenomenon in which ATP functions as a biological hydrotrope of protein aggregates. In this manuscript, by conducting molecular dynamics simulations of three different proteins, Trp-cage, Abeta40 monomer, and Abeta40 dimer at concentrations of ATP (0.1, 0.5 M), which are higher than those at cellular condition (a few mM), Sarkar et al. find that the amphiphilic nature of ATP, arising from its molecular structure consisting of phosphate group (PG), sugar ring, and aromatic base, enables it to interact with proteins in a protein-specific manner and prevents their aggregation and solubilize if they aggregate. The authors also point out that in comparison with NaXS, which is the traditional chemical hydrotrope, ATP is more efficient in solubilizing protein aggregates because of its amphiphilic nature.Trp-cage, featured with hydrophobic core in its native state, is denatured at high ATP concentration. The authors show that the aromatic base group (purine group) of ATP is responsible for inducing the denaturation of helical motif in the native state.For Abeta40, which can be classified as an IDP with charged residues, it is shown that ATP disrupts the salt bridge (D23-K28) required for the stability of beta-turn formation.By showing that ATP can disassemble preformed protein oligomers (Abeta40 dimer), the authors suggest that ATP is "potent enough to disassemble existing protein droplets, maintaining proper cellular homeostasis," and enhancing solubility.Overall, the message of the paper is clear and straightforward to follow. In addition to the previous studies in the literature on this subject. (J. Am. Chem. Soc. 2021, 143, 31, 11982-11993; J. Phys. Chem. B 2022, 126, 42, 8486-8494; J. Phys. Chem. B 2021, 125, 28, 7717-7731; J. Phys. Chem. B 2020, 124, 1, 210-223), the study, which tested using MD simulations whether ATP is a solubilizer of protein aggregates, deserves some attention from the community and is worth publishing.WeaknessMy only major concern is that the simulations were performed at unusually high ATP concentrations (100 and 500 mM of ATP), whereas the real cellular concentration of ATP is 1-5 mM.I was wondering if there is any report on a titration curve of protein aggregates against ATP, and what is the transition mid-point of ATP-induced solubility of protein aggregates. For instance, urea or GdmCl have long been known as the non-specific denaturants of proteins, and it has been well experimented that their transition mid-points of protein unfolding are in the range of ~(1 - 6) M depending on the proteins.The authors responded to my comment on ATP concentration that because of the computational issue in all-atom simulations, they had no option but to employ mM-protein concentrations instead of micromolar concentrations, thus requiring 1000-folds higher ATP concentration, which is at least in accordance with the protein/ATP stoichiometry. However, I believe this is an issue common to all the researchers conducting MD simulations. Even if the system is in the same stoichiometric ratio, it is never clear to me (is it still dilute enough?) whether the mechanism of solubilization of aggregate at 1000 fold higher concentration of ATP remains identical to the actual process.

Thank you for your thoughtful feedback and for recognizing the value of our study. We appreciate your detailed review and the constructive comments you have provided.

We appreciate your understanding of the inherent limitations in MD simulations. The use of higher ATP concentrations in our simulations stems from the computational challenges of all-atom MD simulations. Due to the practical constraints of simulating micromolar protein concentrations in atomistic detail, we employed millimolar protein concentrations, which necessitated the use of ATP concentrations that are proportionally higher to maintain appropriate stoichiometry between ATP and proteins.

We fully agree with your point that this is a common issue faced by researchers in the MD simulation community. While it is challenging to directly replicate physiological ATP concentrations in atomistic simulations, we believe that our approach still captures the fundamental interactions between ATP and proteins. In particular, our focus was on the relative behaviors and mechanistic insights, rather than absolute concentration effects. We based our choice of ATP concentration on maintaining stoichiometric ratios with the protein concentration to ensure that the molecular mechanisms observed remain relevant. We hope our clarification addresses your concerns.

We would like to share that in an ongoing study focused on the role of ATP in influencing the liquid-liquid phase separation behavior of several intrinsically disordered proteins, we are employing a coarse-grained model. This approach allows us to maintain ATP concentrations within physiologically relevant ranges, as simulating micromolar protein concentrations becomes computationally feasible with this method. We believe that this complementary work will provide additional insights into the behavior of ATP at concentrations more reflective of cellular conditions and further validate the findings from our current study.

We would also like to emphasize that the complementary experiments presented in this study were conducted at physiologically relevant concentrations for both protein and ATP. The experimental results are in strong agreement with our computational findings, supporting the hypothesis that the mechanisms observed in the simulations closely reflect the actual biological process.

--—-

The following is the authors’ response to the original reviews.

**Reviewer #1 (Public Review):**
Summary:This work combines molecular dynamics (MD) simulations along with experimental elucidation of the efficacy of ATP as a biological hydrotrope. While ATP is broadly known as the energy currency, it has also been suggested to modulate the stability of biomolecules and their aggregation propensity. In the computational part of the work, the authors demonstrate that ATP increases the population of the more expanded conformations (higher radius of gyration) in both a soluble folded mini-protein Trp-cage and an intrinsically disordered protein (IDP) Aβ40. Furthermore, ATP is shown to destabilise the pre-formed fibrillar structures using both simulation and experimental data (ThT assay and TEM images). They have also suggested that the biological hydrotrope ATP has significantly higher efficacy as compared to the commonly used chemical hydrotrope sodium xylene sulfonate (NaXS).Strengths:This work presents a comprehensive and compelling investigation of the effect of ATP on the conformational population of two types of proteins: globular/folded and IDP. The role of ATP as an "aggregate solubilizer" of pre-formed fibrils has been demonstrated using both simulation and experiments. They also elucidate the mechanism of action of ATP as a multi-purpose solubilizer in a protein-specific manner. Depending on the protein, it can interact through electrostatic interactions (for predominantly charged IDPs like Aβ40), or primarily van der Waals' interactions through (for Trp-Cage).Weaknesses:The data presented by the authors are sound and adequately support the conclusions drawn by the authors. However, there are a few points that could be discussed or elucidated further to broaden the scope of the conclusions drawn in this work as discussed below:(i) The concentration of ATP used in the simulations is significantly higher (500 mM) as compared to those used in the experiments (6-20 mM) or cellular cytoplasm (~5 mM as mentioned by the authors). Since the authors mention already known concentration dependence of the effect of ATP, it is worth clarifying the possible limitations and implications of the high ATP concentrations in the simulations.

We thank the reviewer for their concern regarding the ATP concentration used in our simulation. The reviewer correctly noted our statement about cellular ATP concentrations being in the range of a few millimolar. We would like to highlight that, in a cellular environment, millimolar ATP concentrations coexist with micromolar protein concentrations in the aqueous phase [1].

In our study, we focused on the impact of ATP on protein conformational dynamics, primarily simulating a protein monomer within the simulation box. If one was required to maintain a micromolar protein concentration (e.g., 20 μM [1]) for a monomeric protein, a MD simulation box of significant dimensions (~44x44x44 nm³) would be required, which is computationally challenging to simulate at an atomistic resolution due to the excessive computational cost and time. We had observed a severe reduction of performance of simulation (with Gromacs software of version 2018.6) of more than 150 times for the 20 μM Aβ40 protein in 20 mM ATP solution containing 50 mM NaCl salt which is comprised in the simulation box of ~ 44x44x44 nm³ in comparison to the current simulation set up we have employed in our study.

To ensure computational efficiency, we employed a simulation protocol that would maintain the cellular protein/ATP stoichiometry. Similar to the stoichiometry in the cellular environment (i.e., micromolar protein : millimolar ATP ~ 103), our simulations maintained a consistent ratio (i.e., millimolar protein : molar ATP ~ 103). This approach allowed us to use a smaller simulation box while preserving the relevant stoichiometry, enabling us to leverage data within a realistic timeframe.

Based on the reviewer comment we have included the explanation in the revised manuscript as “In this study, we opted to maintain the ATP stoichiometry consistent with biological conditions and previous in vitro experiments. Instead of keeping the protein concentration within the micromolar range and ATP concentration at the millimolar level, we chose this approach to avoid the need for an extremely large simulation box, which would greatly reduce computational efficiency by more than 150-fold.” (page 4).

However, during our experimental measurements we have maintained micromolar concentration of protein and ATP concentration in the millimolar range, which lies consistent with the former in vitro experimental studies [1].

It seems ATP can stabilise the proteins at low concentrations, but the current work does not address this possible effect. It would be interesting to see whether the effect of ATP on globular proteins and IDPs remains similar even at lower ATP concentrations.

We thank the reviewer for raising this point. We would like to refer you to the Discussion and Conclusion sections of our manuscript (on page 18), where we have noted ATP’s concentration-dependent actions on protein homeostasis, incorporating insights from previous literature as well: “In our literature survey of ATP's concentration-dependent actions, as detailed in the Introduction section, we observed a dual role where ATP induces protein liquid-liquid phase separation at lower concentrations and promotes protein disaggregation at higher concentrations [2–4]. These versatile functions emphasize ATP's pivotal role in maintaining a delicate balance between protein stability (at low ATP concentrations) and solubility (at high ATP concentrations) for effective proteostasis within cells. Notably, ATP-mediated stabilization primarily targets soluble proteins, particularly those with ATP-binding motifs, while ATP-driven biomolecular solubilization is observed for insoluble proteins, typically lacking ATP-binding motifs.”. We explain that ATP stabilizes proteins at lower concentrations, primarily targeting those with ATP-binding motifs, as illustrated by a sequence-dependent analysis. Since the proteins we studied (Trp-cage and Aβ40) do not contain any ATP-binding motifs, ATP-guided protein stabilization is not expected for these proteins. Additionally, we presented a set of simulations for Trp-cage with a comparatively lower concentration of ATP (see Figure 2), which also suggests

ATP-driven protein chain elongation. Thus, we believe that ATP’s effect on globular proteins and intrinsically disordered proteins (IDPs) lacking ATP-binding motifs would remain similar at lower ATP concentrations.”

(ii) The authors make a somewhat ambitious statement that the role of ATP as a solubilizer of pre-formed fibrils could be used as a therapeutic strategy in protein aggregation-related diseases. However, it is not clear how it would be so since ATP is a promiscuous substrate in several biochemical processes and any additional administration of ATP beyond normal cellular concentration (~5 mM) could be detrimental.

The authors thank the reviewer for this comment. In conjunction with earlier studies on the non-energetic effects of ATP, our study underscores ATP’s anti-aggregation properties and its ability to dissolve preformed aggregates, thereby maintaining regular protein homeostasis within cells and inhibiting protein aggregation-related diseases. Consequently, ATP has been proposed as a probable therapeutic agent in multiple previous reports [5–8]. Patel et al. also noted that as ATP levels decrease with age, this can lead to increased protein aggregation and neurodegenerative decline [1]. Therefore, the problem of excessive protein aggregation in cells may be linked to the reduction of ATP levels with aging [1,8–12]. In such circumstances, authors hypothesize introducing ATP as part of a therapeutic treatment might address the issue of excessive protein aggregation and neurodegenerative diseases.

(iii) A natural question arises about what is so special about ATP as a solubilizer. The authors have also asked this question but in a limited scope of comparing to a commonly used chemical hydrotrope NaXS. However, a bigger question would be what kind of chemical/physical features make ATP special? For example, (i) if the amphiphilic property is important, what about some standard surfactants? (ii) how would ATP compare to other nucleotides like ADP or GTP? It might be useful to explore such questions in the future to further establish the special role of ATP in this regard.

We thank the reviewer for recognizing the significance and value of our exploration into the unique properties of ATP as a solubilizer. In response to the reviewer’s comment regarding the specific features that make ATP special, we would like to emphasize our analysis of ATP's region-specific interactions with biomolecules. ATP's unique structure, comprising three distinct moieties- a larger hydrophobic aromatic base, a hydrophilic sugar moiety, and a highly negatively charged phosphate group, enables it to perform multiple modes of interactions, including hydrophobic, hydrogen bonding, and electrostatic interactions with proteins. This combination of interactions leads to its pronounced effect in a protein-specific manner. We believe that, together with its amphiphilic property, the specific chemical structure of ATP makes it an efficient solubilizer. A previous study by Patel et al. demonstrated the efficiency of ATP as a biological hydrotrope compared to other classical chemical hydrotropes (NaXS and NaTO). Our current study further rationalizes ATP’s efficiency through its effective interactions with biomolecules, driven by the chemically distinct parts of the ATP molecule.

Regarding the reviewer’s point about comparing ATP as a hydrotrope with standard surfactants, we would like to add that typically, hydrotropes are amphiphilic molecules that differ from classical surfactants due to their low cooperativity of aggregation and their effectiveness at molar concentrations. Hydrotropes tend to preferentially accumulate non stoichiometrically around the solute, and their aggregation depends on the presence of solute molecules. Unlike surfactants, hydrotropes do not form any well-defined superstructure on their own.

In response to the reviewer’s comment on comparing ATP’s effect with other nucleotides like ADP and GTP, we would like to highlight that previous studies have shown GTP to dissolve protein droplets (FUS) with similar efficiency to ATP. However, in cells, the concentration of GTP is much lower than that of ATP, resulting in negligible effects on the solubilization of liquid compartments in vivo. Conversely, ADP and AMP exhibited comparatively lower efficiency in dissolving protein condensates, suggesting the triphosphate moiety plays a considerable role in protein condensate dissolution. Additionally, only TP-Mg had a negligible effect on protein drop dissolution, indicating that the charge density in the ionic ATP side chain alone is insufficient for dissolving protein drops. Together, these findings highlight the efficiency of ATP as a protein aggregate solubilizer, which stems from its specific chemical structure and not merely its amphiphilicity.

According to the suggestion of the reviewer we have included the discussion in the revised manuscript as “Comparing the effects of ATP with other nucleotides such as ADP and GTP, we emphasize that previous studies have demonstrated GTP can dissolve protein droplets (such as FUS) with efficiency comparable to ATP. However, in vivo, the concentration of GTP is significantly lower than that of ATP, resulting in negligible impact on the solubilization of liquid compartments. In contrast, ADP and AMP show much lower efficiency in dissolving protein condensates, indicating the critical role of the triphosphate moiety in protein condensate dissolution. Furthermore, only TP-Mg exhibited a negligible effect on protein droplet dissolution, suggesting that the charge density in the ionic ATP side chain alone is insufficient for this process. These findings underscore ATP's superior efficacy as a protein aggregate solubilizer, attributed to its specific chemical structure rather than merely its amphiphilicity.” (page 15).

(iv) In Figure 2F, it seems that in the presence of 0.5 M ATP, the Rg increases (as expected), but the number of native contacts remains almost similar. The reduction in the number of native contacts at higher ATP concentrations is not as dramatic as the increase in Rg. This is somewhat counterintuitive and should be looked into. Normally one would expect a monotonous reduction in the number of native contacts as the protein unfolds (increase in Rg).

We appreciate the reviewer’s insightful comment. As noted, the presence of 0.5 M ATP results in an increase in the protein’s radius of gyration (Rg) and a decrease in native contacts, indicating that ATP promotes protein chain extension. However, the extent of the changes in Rg and native contacts are not identical. It is important to recognize that even the disruption of a few native contacts can significantly impact protein folding, leading to considerable protein chain extension. Therefore, it is not necessary for the extent of variation in Rg and native contacts to be similar. The appropriate measure is whether the alterations in these two variables are consistent with each other, such that an increase in Rg is accompanied by a decrease in native contacts, and vice versa.

**Reviewer #1 (Recommendations For The Authors):**
(i) There are several references repeated multiple times, e.g. (a) 1, 9, 14, (b) 25, 29, 31, 33. There are more such examples and the authors should fix these.

We thank the reviewer for pointing this out. We have addressed the issue in the updated manuscript.

(ii) Specific Gromacs version should be mentioned rather than 20xx.

In the updated manuscript we have mentioned the particular version of Gromacs software (2018.6) we have employed for our simulation.

**Reviewer #2 (Public Review):**
In this work, Sarkar et al. investigated the potential ability of adenosine triphosphate (ATP) as a solubilizer of protein aggregates by combining MD simulations and ThT/TEM experiments. They explored how ATP influences the conformational behaviors of Trp-cage and β-amyloid Aβ40 proteins. Currently, there are no experiments in the literature supporting their simulation results of ATP on Trp-cage. The simulation protocol employed for the Aβ40 monomer system is conventional MD simulation, while REMD simulation (an enhanced sampling method) is used for the Aβ monomer + ATP system. It is not clear whether the conformational difference is caused by ATP or by the different simulation methods used.

We thank the reviewer for raising this point. First we note that for Trp-cage, the simulation methods employed in presence and absence of ATP were identical (REMD simulation) and the difference in the free energy surfaces due to introduction of ATP in the solution were evident.

Nonetheless to address referee’s point if the difference in simulation method employed for generating the 2D free energy landscape in absence and presence of ATP would have introduced the observed difference, we had undertaken the initiative of carrying out a fresh set of REMD simulations with Aβ40 in neat water, followed by adaptive sampling simulation. As shown below in Author response image 1, the free energy profiles obtained from conventional MD simulation (using DESRES trajectory) as well as those obtained via REMD simulations for the same system (in neat water) are qualitatively similar. The free energy profiles obtained in presence of ATP are significantly different from that of neat water, irrespective of the simulation method. This confirms the simulation’s observation of ATP driven alteration of protein conformation.

**Author response image 1. sa3fig1:** Image represents the 2D free energy profile for Aβ40 monomer in absence of ATP, obtained through A. conventional MD and B. REMD simulation followed by adaptive sampling simulation.

In the revised manuscript we have included the discussion as “To verify that the effect of ATP on conformational landscape is not an artifact of difference in sampling method (long conventional MD in absence of ATP versus REMD in presence of ATP), we repeated the conformational sampling in absence of ATP via employing REMD, augmented by adaptive sampling (figure S4). We find that the free energy map remains qualitatively similar (figure 4A and S4) irrespective the sampling technique. Comparison of 2D free energy map obtained from REMD simulation in absence of ATP (figure S4) with the one obtained in presence of ATP (figure 4B) also indicates ATP driven protein chain elongation.” on page 7 and updated the method section as “To test the robustness we have also estimated the 2D free energy profile of Aβ40 in absence of ATP by performing a similar REMD simulation followed by adaptive sampling simulation following the similar protocol described above.” on page 20.

ThT/TEM experiments should be performed on Aβ40 fibrils rather than on Aβ(16-22) aggregates. Moreover, to elucidate their experimental results that ATP can dissolve preformed Aβ fibrils, the authors need to study the influence of ATP on Aβ fibrils instead of on Aβ dimer in their MD simulations. The novelty of this study is limited. The role of ATP in inhibiting Aβ fibril formation and dissolving preformed Aβ fibrils has been reported in previous experimental and computational studies (Journal of Alzheimer's Disease, 2014, 41: 561; Science 2017, 2017, 356, 753-756 J. Phys. Chem. B 2019, 123, 9922−9933; Scientific Reports, 2024, 14: 8134). However, most of those papers are not discussed in this manuscript. Additionally, some details of MD simulations and data analysis are missing in the manuscript, including the initial structures of all the simulations, the method for free energy calculation, the dielectric constant used, etc.

We thank the reviewer for pointing out additional papers on ATP that were not discussed in the original manuscript. While some of the suggested papers were already cited (Science 2017, 356, 753-756), we had initially excluded the others as we did not find them directly relevant to our focus. However, in this revised version, we have included those references (on page 17 and 18).

Through a thorough literature review, including the papers suggested by the reviewer, we maintain that our article is novel in its investigation of ATP's role in the protein conformational landscape and its correlation with anti-aggregation effects. While previous reports emphasize ATP's role in inhibiting protein aggregation, our work connects these findings by highlighting ATP's influence starting at the monomeric level, thereby preventing proteins from becoming aggregation-prone.

In the revised manuscript, we have included this justification as “While previous reports emphasize ATP's role in inhibiting protein aggregation, our work connects these findings by highlighting ATP's influence starting at the monomeric level, thereby preventing proteins from becoming aggregation-prone.” on page 18.

Regarding the reviewer's concern on the details of MD simulations, we would like to mention that method part of the current article provides an elaborate explanation of the simulation set up and characterization (on page 19-21). Regarding the reviewer's comment on dielectric constant, we would like to emphasize that here we have performed simulation considering explicit presence of solvent (water molecules), which by default takes into account dielectric constants (unlike many approximate continuum modelling approaches).

**Reviewer #2 (Recommendations For The Authors):**
(1) The convergence of simulations needs to be verified prior to data analysis.

We thank the reviewer for this suggestion. We have assessed the convergence of the simulations and represented the respective plots in Author response image 2.

**Author response image 2. sa3fig2:** The time profile of temperature (a, c, e and g) and energies i. e. kinetic energy, potential energy and total energy (b, d, f and h) are being represented for Trp-cage in absence (a-b) and presence of 0.5 MATP (c-d) and Aβ40 protein in absence (e-f) and presence of 0.5 M ATP (g-h).

(2) "The precedent experiments investigating protein aggregation in the presence of ATP, had been performed by maintaining the ATP:protein stoichiometric ratio in the range of 0.1x10x3 to 1.6x10x3. Likewise, in our simulation with Trp-cage, the ATP:protein ratio of 0.02x10x3 was maintained.". Clearly, there is a big difference between the ATP:protein ratio in the MD simulations and that in the precedent experiments.

We thank the reviewer for raising this point. We would like to clarify that for unstructured proteins, including Aβ40, the ATP stoichiometry [1] ranged from 0.1 × 10³ to 1.6 × 10³. In our study, we have maintained the ATP stoichiometry at 0.1 × 10³ for the disordered protein Aβ40. For structured globular mini-protein like Trp-cage, a lower concentration of 0.02 × 10³ was used, consistent with other studies investigating the effects of ATP on globular proteins such as ubiquitin, lysozyme, and malate dehydrogenase, where the ATP stoichiometry ranged [13] from 0.01 × 10³ to 0.03 × 10³.

In the revised manuscript we have clearly mentioned the point as “The precedent studies reporting the effect of ATP on structured proteins, had been performed by maintaining ATP:protein stoichiometric ratio in the range of 0.01x103 to 0.03x103. Likewise, in our simulation with Trp-cage, the ATP:protein ratio of 0.02x103 was maintained. ” in page 4 and “The former experiments investigating protein (unstructured) aggregation in presence of ATP, had been performed by maintaining ATP:protein stoichiometric ratio in the range of 0.1x103 to 1.6x103, similarly we have also maintained ATP/protein stoichiometry 0.1x103 in our investigation ATP’s effect on disordered protein Aβ40.” in page 7.

However, during our experimental measurements we have maintained micromolar concentration of protein and ATP concentration in the millimolar range, which lies consistent with the former in vitro experimental studies [1].

(3) The snapshots in Figure 2G show that in the absence of ATP, the Trp-cage monomer exhibits only minor conformational changes compared to the NMR structure (PDB: 1L2Y). However, the native contact number of the Trp-cage monomer (~18, Figure 2C) is much smaller than the total contact number (~160, Figure 2B). The authors are suggested to explain this unexpectedly large difference.

The authors thank the reviewer for his/her concern related to the values of native contact and the total number of contacts of the protein Trp-cage. The author would like to highlight that the estimation of total number of contacts involves the cumulative number of intra-protein contacts which calculates when the two atoms of the protein’s come within the cut-off distance (0.8 nm). Whereas native contact only considers the key contacts of the protein between the side chains of two amino acids that are not adjacent in the amino acid sequence.

(4) The authors are suggested to calculate the contact numbers of each residue with different parts of ATP (phosphate group, base, and sugar moiety), which will help to reveal the key interactions between ATP and proteins.

The authors thank the reviewer for this comment. According to the suggestion we have calculated the contact probability of each residue of protein with ATP as depicted in Author response image 3 and 4 for Trp-cage and Aβ40 respectively.

**Author response image 3. sa3fig3:** The figure shows the residue wise contact probability of protein Trp-cage with ATP.

**Author response image 4. sa3fig4:** The image shows the residue wise contact probability of Aβ40 protein with ATP.

For detailed interaction of ATP’s region-specific interactions with proteins, the authors would like to refer to the calculation of the preferential binding coefficient and interaction energies as depicted in Figure 3 for Trp-cage (in page 6) and in Figure 5 and 8 for Aβ40 protein. These figures illustrate well the mode of protein interaction with the chemically divergent regions of ATP and also illuminates ATP’s interaction with different parts of the proteins as well.

(5) The authors claimed that "coulombic interaction of ATP with protein predominates in Aβ40 (Figure 5 H)" (Page 10). However, the preferential interaction coefficient in Figure 5G shows that the curve of the phosphate group lies below the other two curves when distance < 1 nm, indicating the relatively weak interactions between the phosphate group and Aβ40. This seems to be in conflict with the results of energy calculation (Figure 5H).

We thank the reviewer for raising this point. The author would like to emphasize that ATP, with its large and highly charged phosphate group, is highly likely to interact with intrinsically disordered proteins (IDPs) primarily through electrostatic interactions due to their significant charge content. In Figure 5G, it is evident that the preferential binding coefficient reaches a notably high value, indicating strong interaction between the protein and the charged phosphate group of ATP. To address the reviewer's concern regarding the curve showing the highest interaction value only after 1 nm, we would like to highlight the nature of long-range electrostatic potential, which is active in the range of approximately 1-1.2 nm [14–16]. Furthermore, Figure 5H confirms that the electrostatic interaction between the protein and ATP is favorable and predominates over the Lennard-Jones (LJ) interaction.

(6) There are several issues with citations. For example, references 2, 5, 24, 28, 32, 45. 49 and 53 are the same paper, references 1, 7, and 14 are the same paper, references 12, 15, and 46 are the same paper, and many more. In addition, the title of reference 12/15 is "ATP Controls the Aggregation of Aβ16-22 Peptides" instead of "ATP Controls the Aggregation of Aβ Peptides".

We thank the reviewer for pointing this out. We have addressed the issue in the updated manuscript.

(7) References 19 and 20 are cited in the context of "As a potential function of the excess ATP concentration within the cell, a substantial influence on cellular protein homeostasis is observed, particularly in preventing protein aggregation (14-21)" (Page 2). However, there is no mention of "ATP" in ref. 19 and 20.

Thank you to the reviewer for identifying this mistake. We have corrected the issue in the revised manuscript.

(8) On page 22: "To perform all the molecular dynamics (MD) simulations GROMACS software of version 20xx software was utilized". Please provide the version of GROMACS software used in this study.

In the updated manuscript, we have specified the particular version of Gromacs software (2018.6) used for our simulations. (see revised manuscript page 19)

(9) In Figure 8J, the time-dependent distance of Aβ40 dimer without ATP needs to be provided as a comparison.

We thank the reviewer for this comment. In the revised manuscript we have updated the calculation of distance between the Aβ40 protein chains both in absence and presence of ATP as well as “The probability distribution (Figure 8J) illustrates that, in the presence of ATP, the two protein chains, initially part of the dimer, become prone to be moved away from each other.” (page 15).

(10) The authors should compare ATP-Aβ interactions with NaXS-Aβ interactions to understand why ATP is more efficient than NaXS in inhibiting interprotein interactions.

The authors thank the reviewer for the concern regarding the ATP-Aβ40 interaction compared to the NaXS-Aβ40 interaction. We would like to highlight our results (Figure 5G and H) which demonstrate the dominance of Coulombic interactions (over LJ interactions) of ATP with the protein. Based on this, we compared the Coulombic interaction energy of ATP and NaXS with the protein Aβ40, as depicted in Figure 9I. We observed that ATP-protein electrostatic interactions occur more favorably than those with NaXS, leading to better action of ATP over NaXS. The favorable electrostatic interaction of ATP with the protein, compared to NaXS, is evident because ATP possesses a large and highly charged triphosphate group that can strongly interact with the protein, whereas NaXS contains a very small sulfonate group with much less charge. Therefore, due to the favorable Coulombic interaction of ATP with the protein over NaXS, ATP acts more efficiently as a hydrotrope. In the revised manuscript we have highlighted the term “Coulombic interaction” in the main text and in the figure caption (Figure 9) as well (in page 15 and 16 of the revised manuscript respectively).

(11) The word "sollubilizer" in the Abstract is a typo.

We thank the reviewer for pointing this out. We have made the necessary corrections in the revised manuscript.

(12) What does "ATP-Mg2+" mean in the manuscript?

ATP, being polyanionic and possessing a potentially chelating polyphosphate group, binds metal cations with high affinity and hence biologically it occurs to be complexed with the equivalent number of Mg2+ in the form of ATP-Mg [17–19]. Similarly multiple former studies utilized ATP-Mg in their investigations [1,20–22].

**Reviewer #3 (Public Review):**
Summary:Since its first experimental report in 2017 (Patel et al. Science 2017), there have been several studies on the phenomenon in which ATP functions as a biological hydrotrope of protein aggregates. In this manuscript, by conducting molecular dynamics simulations of three different proteins, Trp-cage, Abeta40 monomer, and Abeta40 dimer at a high concentration of ATP (0.1, 0.5 M), Sarkar et al. find that the amphiphilic nature of ATP, arising from its molecular structure consisting of phosphate group (PG), sugar ring, and aromatic base, enables it to interact with proteins in a protein-specific manner and prevents their aggregation and solubilize if they aggregate. The authors also point out that in comparison with NaXS, which is the traditional chemical hydrotrope, ATP is more efficient in solubilizing protein aggregates because of its amphiphilic nature.Trp-cage, featured with a hydrophobic core in its native state, is denatured at high ATP concentration. The authors show that the aromatic base group (purine group) of ATP is responsible for inducing the denaturation of helical motifs in the native state.For Abeta40, which can be classified as an IDP with charged residues, it is shown that ATP disrupts the salt bridge (D23-K28) required for the stability of beta-turn formation.By showing that ATP can disassemble preformed protein oligomers (Abeta40 dimer), the authors argue that ATP is "potent enough to disassemble existing protein droplets, maintaining proper cellular homeostasis," and enhancing solubility.Overall, the message of the paper is clear and straightforward to follow. I did not follow all the literature, but I see in the literature search, that there are several studies on this subject. (J. Am. Chem. Soc. 2021, 143, 31, 11982-11993; J. Phys. Chem. B 2022, 126, 42, 8486-8494; J. Phys. Chem. B 2021, 125, 28, 7717-7731; J. Phys. Chem. B 2020, 124, 1, 210-223).If this study is indeed the first one to test using MD simulations whether ATP is a solubilizer of protein aggregates, it may deserve some attention from the community. But, the authors should definitely discuss the content of existing studies, and make it explicit what is new in this study.Strengths:The authors showed that due to its amphiphilic nature, ATP can interact with different proteins in a protein-specific manner, a. finding more general and specific than merely calling ATP a biological hydrotrope.Weaknesses:(1) My only major concern is that the simulations were performed at unusually high ATP concentrations (100 and 500 mM of ATP), whereas the real cellular concentration of ATP is 1-5 mM. Even if ATP is a good solubilizer of protein aggregates, the actual concentration should matter. I was wondering if there is a previous report on a titration curve of protein aggregates against ATP, and what is the transition mid-point of ATP-induced solubility of protein aggregates.For instance, urea or GdmCl have long been known as the non-specific denaturants of proteins, and it has been well experimented that their transition mid-point of protein unfolding is ~(1 - 6) M depending on the proteins.

We thank the reviewer for their concern regarding the ATP concentration used in our simulation. The reviewer correctly noted our statement about cellular ATP concentrations being in the range of a few millimolar. We would like to highlight that, in a cellular environment, millimolar ATP concentrations coexist with micromolar protein concentrations in the aqueous phase.

In our study, we focused on the impact of ATP on protein conformational dynamics, primarily simulating a protein monomer within the simulation box. To maintain a micromolar protein concentration (e.g., 20 μM [1]) for a monomeric protein, a simulation box of significant dimensions (~44x44x44 nm³) would be required. This size would be computationally challenging to simulate at an atomistic resolution due to the excessive computational cost and time.

To ensure computational efficiency, we employed millimolar protein concentrations instead of micromolar, thus requiring a higher ATP concentration to maintain the cellular protein stoichiometry. Similar to the stoichiometry in the cellular environment (i.e., micromolar protein : millimolar ATP ~ 103), our simulations maintained a consistent ratio (i.e., millimolar protein : molar ATP ~ 103). This approach allowed us to use a smaller simulation box while preserving the relevant stoichiometry, enabling us to leverage data within a realistic timeframe.

Based on the reviewer comment we have included the explanation in the revised manuscript as “In this study, we opted to maintain the ATP stoichiometry consistent with biological conditions and previous in vitro experiments. Instead of keeping the protein concentration within the micromolar range and ATP concentration at the millimolar level, we chose this approach to avoid the need for an extremely large simulation box, which would greatly reduce computational efficiency by more than 150-fold.” (page 4).

However, during our experimental measurements we have maintained micromolar concentration of protein and ATP concentration in the millimolar range, which lies consistent with the former in vitro experimental studies [1]

(2) The sentence "... a clear shift of relative population of Abeta40 conformational subensemble towards a basin with higher Rg and lower number of contacts in the presence of ATP" is not a precise description of Figures 4A and 4B. It is not clear from the figures whether the Rg of Abeta40 is increased when Abeta40 is subject to ATP. The authors should give a more precise description of what is observed in the result from their simulations or consider a better-order parameter to describe the change in molecular structure.

We thank the reviewer for this comment. Figure 4A and 4B depicting the 2D free energy profile of the Aβ40 protein with respect to Rg and total number contacts are presented to pinpoint the alteration of protein conformational landscape in influence of ATP. To further elucidate ATP driven protein conformational alteration, the overlaid snapshots corresponding to absence and presence of ATP were also provided. Together the author believes that the descriptions of Figures 4A and 4B in the article are appropriate and effectively incorporate the analysis provided in the article.

In addition, the disruption of beta-sheet from Figure 4E to 4F is not very clear. The authors may want to use an arrow to indicate the region of the contact map associated with this change.

In the revised manuscript the authors have highlighted the region of the contact map associated with the changes in the beta-sheet propensity with an arrow for each of the plots.

Although the full atomistic simulations were carried out, the analyses demonstrated in this study are a bit rudimentary and coarse-grained (e.g, Rg is a rather poor order parameter to discuss dynamics involved in proteins). The authors could go beyond and say more about how ATP interacts with proteins and disrupts the stable configurations.

We thank the reviewer for this comment. We understand the reviewer's concern regarding the choice of the order parameter (Rg), which has been a topic of long-standing debate. However, we would like to note that in the current study, we employed Rg based on recent investigations by Dr. D. E. Shaw Research group [23] (specifically concerning the protein Aβ40 and the Charmm36m force field), which reported an almost negligible Rg penalty compared to experimental values. The experiments characterizing IDPs utilize Rg as a choice of metric. We also would like to highlight that previous investigations of our group have done careful benchmarking of several features of proteins as well as IDPs using both linear and artificial neural network based dimension reduction techniques and have demonstrated that Rg, in combination with fraction of native contact serves as optimum features [24,25]. Therefore, we believed that Rg would be a suitable order parameter for analyzing the structural behavior of this protein. Additionally, we have also analyzed other relevant characteristics, including the total number of contacts, residue-wise protein contact map, percentage of secondary structure, solvent-accessible surface area, and distances between key interacting residues, to provide a comprehensive understanding.

The justification of our choice of collective variable has been discussed in the revised manuscript as “Since multiple previous studies has reported benchmarking of several features of proteins as well as IDPs using both linear and artificial neural network based dimension reduction techniques and have demonstrated that Rg, in combination with fraction of native contact serves as optimum features, we have chosen these two metrics for developing the 2D free energy profile.” on page 4.

(3) Although the amphiphilic character of ATP is highlighted, a similar comment can be made as to GTP. Is GTP, whose cellular concentration is ~0.5 mM, also a good solubilizer of protein aggregates? If not, why? Please comment.

In response to the reviewer’s comment on comparing ATP’s effect with other nucleotides GTP, we would like to highlight that previous studies have shown GTP’s ability to dissolve protein droplets (FUS) with similar efficiency to ATP [1,26]. However, in cells, the concentration of GTP is much lower than that of ATP, resulting in negligible effects on the solubilization of liquid compartments in vivo [1].

According to the suggestion of the reviewer we have included the discussion in the revised manuscript as “Comparing the effects of ATP with other nucleotides such as ADP and GTP, we emphasize that previous studies have demonstrated GTP can dissolve protein droplets (such as FUS) with efficiency comparable to ATP. However, in vivo, the concentration of GTP is significantly lower than that of ATP, resulting in negligible impact on the solubilization of liquid compartments. In contrast, ADP and AMP show much lower efficiency in dissolving protein condensates, indicating the critical role of the triphosphate moiety in protein condensate dissolution. Furthermore, only TP-Mg exhibited a negligible effect on protein droplet dissolution, suggesting that the charge density in the ionic ATP side chain alone is insufficient for this process. These findings underscore ATP's superior efficacy as a protein aggregate solubilizer, attributed to its specific chemical structure rather than merely its amphiphilicity.” (page 15).

**Reviewer #3 (Recommendations For The Authors):**
Spell-check should be carried out throughout the manuscript. e.g., sollubilizer, sollubilizing, ...

We thank the reviewer for pointing this out. We have made the necessary corrections in the revised manuscript.

The reference section should be properly organized. There are multiple repetitions of references (e.g., references 28, 30, 32 are the same reference). I see many instances of this.

We thank the reviewer for pointing this out. We have addressed the issue in the updated manuscript.

References:

(1) Patel, A.; Malinovska, L.; Saha, S.; Wang, J.; Alberti, S.; Krishnan, Y.; Hyman, A. A. ATP as a Biological Hydrotrope. Science 2017, 356 (6339), 753–756.

(2) Ren, C.-L.; Shan, Y.; Zhang, P.; Ding, H.-M.; Ma, Y.-Q. Uncovering the Molecular Mechanism for Dual Effect of ATP on Phase Separation in FUS Solution. Sci Adv 2022, 8 (37), eabo7885.

(3) Song, J. Adenosine Triphosphate Energy-Independently Controls Protein Homeostasis with Unique Structure and Diverse Mechanisms. Protein Sci. 2021, 30 (7), 1277–1293.

(4) Liu, F.; Wang, J. ATP Acts as a Hydrotrope to Regulate the Phase Separation of NBDY Clusters. JACS Au 2023, 3 (9), 2578–2585.

(5) Chu, X.-Y.; Xu, Y.-Y.; Tong, X.-Y.; Wang, G.; Zhang, H.-Y. The Legend of ATP: From Origin of Life to Precision Medicine. Metabolites 2022, 12 (5). https://doi.org/10.3390/metabo12050461.

(6) Tian, Z.; Qian, F. Adenosine Triphosphate-Induced Rapid Liquid-Liquid Phase Separation of a Model IgG1 mAb. Mol. Pharm. 2021, 18 (1), 267–274.

(7) Wang, B.; Zhang, L.; Dai, T.; Qin, Z.; Lu, H.; Zhang, L.; Zhou, F. Liquid-Liquid Phase Separation in Human Health and Diseases. Signal Transduct Target Ther 2021, 6 (1), 290.

(8) Alberti, S.; Dormann, D. Liquid-Liquid Phase Separation in Disease. Annu. Rev. Genet. 2019, 53, 171–194.

(9) Nair, K. S. Aging Muscle. Am. J. Clin. Nutr. 2005, 81 (5), 953–963.

(10) Recharging Mitochondrial Batteries in Old Eyes. Near Infra-Red Increases ATP. Exp. Eye Res. 2014, 122, 50–53.

(11) Goldberg, J.; Currais, A.; Prior, M.; Fischer, W.; Chiruta, C.; Ratliff, E.; Daugherty, D.; Dargusch, R.; Finley, K.; Esparza-Moltó, P. B.; Cuezva, J. M.; Maher, P.; Petrascheck, M.; Schubert, D. The Mitochondrial ATP Synthase Is a Shared Drug Target for Aging and Dementia. Aging Cell 2018, 17 (2). https://doi.org/10.1111/acel.12715.

(12) Kagawa, Y.; Hamamoto, T.; Endo, H.; Ichida, M.; Shibui, H.; Hayakawa, M. Genes of Human ATP Synthase: Their Roles in Physiology and Aging. Biosci. Rep. 1997, 17 (2), 115–146.

(13) Ou, X.; Lao, Y.; Xu, J.; Wutthinitikornkit, Y.; Shi, R.; Chen, X.; Li, J. ATP Can Efficiently Stabilize Protein through a Unique Mechanism. JACS Au 2021, 1 (10), 1766–1777.

(14) Norberg, J.; Nilsson, L. On the Truncation of Long-Range Electrostatic Interactions in DNA. Biophys. J. 2000, 79 (3), 1537–1553.

(15) Pabbathi, A.; Coleman, L.; Godar, S.; Paul, A.; Garlapati, A.; Spencer, M.; Eller, J.; Alper, J. D. Long-Range Electrostatic Interactions Significantly Modulate the Affinity of Dynein for Microtubules. Biophys. J. 2022, 121 (9), 1715–1726.

(16) Sastry, M. Nanoparticle Thin Films: An Approach Based on Self-Assembly. In Handbook of Surfaces and Interfaces of Materials; Elsevier, 2001; pp 87–123.

(17) Wilson, J. E.; Chin, A. Chelation of Divalent Cations by ATP, Studied by Titration Calorimetry. Anal. Biochem. 1991, 193 (1), 16–19.

(18) Storer, A. C.; Cornish-Bowden, A. Concentration of MgATP2- and Other Ions in Solution. Calculation of the True Concentrations of Species Present in Mixtures of Associating Ions. Biochem. J 1976, 159 (1), 1–5.

(19) Garfinkel, L.; Altschuld, R. A.; Garfinkel, D. Magnesium in Cardiac Energy Metabolism. J. Mol. Cell. Cardiol. 1986, 18 (10), 1003–1013.

(20) Hautke, A.; Ebbinghaus, S. The Emerging Role of ATP as a Cosolute for Biomolecular Processes. Biol. Chem. 2023, 404 (10), 897–908.

(21) Pal, S.; Roy, R.; Paul, S. Deciphering the Role of ATP on PHF6 Aggregation. J. Phys. Chem. B 2022, 126 (26), 4761–4775.

(22) Pal, S.; Paul, S. ATP Controls the Aggregation of Aβ Peptides. J. Phys. Chem. B 2020, 124(1), 210–223.

(23) Robustelli, P.; Piana, S.; Shaw, D. E. Developing a Molecular Dynamics Force Field for Both Folded and Disordered Protein States. Proc. Natl. Acad. Sci. U. S. A. 2018, 115 (21), E4758–E4766.

(24) Ahalawat, N.; Mondal, J. Assessment and Optimization of Collective Variables for Protein Conformational Landscape: GB1 -Hairpin as a Case Study. J. Chem. Phys. 2018, 149 (9), 094101.

(25) Menon, S.; Adhikari, S.; Mondal, J. An Integrated Machine Learning Approach Delineates Entropy-Mediated Conformational Modulation of α-Synuclein by Small Molecule, 2024. https://doi.org/10.7554/elife.97709.1.

(26) Pandey, M. P.; Sasidharan, S.; Raghunathan, V. A.; Khandelia, H. Molecular Mechanism of Hydrotropic Properties of GTP and ATP. J. Phys. Chem. B 2022, 126 (42), 8486–8494.